# Novelty is not surprise: Human exploratory and adaptive behavior in sequential decision-making

He A. Xu[1]☯, Alireza Modirshanechi[2,3]☯*, Marco P. Lehmann[2,3], Wulfram Gerstner[2,3‡], Michael H. Herzog[1,2‡]

**1** Laboratory of Psychophysics, School of Life Sciences, Ecole Polytechnique Fédérale de Lausanne (EPFL), Lausanne, Switzerland, **2** Brain-Mind Institute, School of Life Sciences, Ecole Polytechnique Fédérale de Lausanne (EPFL), Lausanne, Switzerland, **3** School of Computer and Communication Sciences, Ecole Polytechnique Fédérale de Lausanne (EPFL), Lausanne, Switzerland

☯ These authors contributed equally to this work.
‡ WG and MHH also contributed equally to this work.
\* alireza.modirshanechi@epfl.ch

**Data Availability Statement:** The experimental data and the code to reproduce all our results are available here: https://github.com/EPFL-LCN/pub-xumodirshanechi2021-PlosCB.

## Abstract

Classic reinforcement learning (RL) theories cannot explain human behavior in the absence of external reward or when the environment changes. Here, we employ a deep sequential decision-making paradigm with sparse reward and abrupt environmental changes. To explain the behavior of human participants in these environments, we show that RL theories need to include surprise and novelty, each with a distinct role. While novelty drives exploration before the first encounter of a reward, surprise increases the rate of learning of a world-model as well as of model-free action-values. Even though the world-model is available for model-based RL, we find that human decisions are dominated by model-free action choices. The world-model is only marginally used for planning, but it is important to detect surprising events. Our theory predicts human action choices with high probability and allows us to dissociate surprise, novelty, and reward in EEG signals.

## Author summary

Humans like to explore their environment: children play with toys, tourists explore touristic sites, and readers start a new book. Exploration is useful to build knowledge about the world in the form of a 'world-model'. However, since the world is complex and changing, the learned world-model is sometimes wrong: if so, the feeling of surprise arises. Here, we distinguish surprise from novelty; we show that humans use surprise as a signal to decide when to adapt their behavior, while they use novelty to decide where and what to explore —to eventually develop an improved world-model. Intuitively, it seems obvious to use world-models to plan future actions. However, we show that in a complex and changing environment where planning needs heavy computations, participants rarely follow an explicit plan and take their actions mainly by shaping habits. Importantly, we show that the main role of their world-model is to signal when to be surprised and, hence, when to

**Funding:** This research was supported by Swiss National Science Foundation No. CRSII2 147636 (Sinergia, MHH and WG) and No. 200020 184615 (WG), and by the European Union Horizon 2020 Framework Program under grant agreement No. 785907 (Human Brain Project, SGA2, MHH and WG). The funders had no role in study design, data collection and analysis, decision to publish, or preparation of the manuscript.

**Competing interests:** The authors have declared that no competing interests exist.

adapt their habits. In summary, our results show how surprise and novelty interact with human reinforcement learning, contribute to human adaptive and exploratory behavior, and correlate with EEG signals.

## Introduction

Humans seek not only explicit rewards such as money or praise [1–8] but also novelty [9, 10], an intrinsic reward-like signal which is linked to curiosity [9–15]. In the theory of reinforcement learning, novelty is considered as a drive for exploration [12, 16–18], and novelty-driven exploratory actions have been interpreted as steps towards building a model of the world ('world-model') which is then used for action planning [19]. A world-model represents implicit knowledge that links actions to observations, such as 'if I open the door to my kitchen, I will see my fridge'.

However, since the world is much more complex than any model of it, there will occasionally be a mismatch between the expectations arising from the model and the actual observation, e.g., when you return from work and the location of the fridge is suddenly empty because your room-mate has sent it off for repair. Such mismatches generate the feeling of surprise, known to manifest in pupil dilation [20] and EEG signals [21–23]. Whereas the reward prediction error (RPE) is a mismatch between the expected reward and the actual reward, surprise is a mismatch between an expected observation and an actual observation. Behavioral experiments [20, 24–27] and theories [27–30] suggest that surprise helps humans to adapt their behavior quickly to changes in the environment, potentially by modulating synaptic plasticity [31–33].

Surprise is fundamentally different from novelty; if you already know that your fridge would be fetched for repairing, the new arrangement of the kitchen without the fridge is novel but not surprising. However, although there is some agreement that novelty and surprise are two separate notions, it has been debated how they can be formally distinguished [34–37], whether they manifest themselves differently in EEG signals [22, 23, 38, 39], and how they influence learning and decision-making [9, 10, 12, 12, 14, 15, 20, 24–26, 37, 40, 41].

In this study, we address three questions: First, how do surprise and novelty influence human reinforcement learning? Second, what is their relative contribution to exploratory and adaptive behavior? And third, can surprise be distinguished from novelty in human behavioral choices and event related potentials (ERP) of the electroencephalogram (EEG)? We show, via a specifically designed deep sequential decision task and a novel hybrid reinforcement learning model, that we can dissociate contributions of surprise from those of novelty and reward in human behavior and ERP.

Our key findings can be summarized in three points: (i) We find that novelty-seeking explains participants' exploratory behavior better than alternative exploration strategies such as seeking surprise or uncertainty [42, 43]; (ii) we observe that participants use their world-model only rarely for action planning and mainly to extract moments of surprise; and importantly, (iii) we show that surprise calculated by the world-model does not only modulate the learning of the world-model [24–26, 29] but also the learning of model-free action-values. In particular, we show that such a modulation is necessary to explain participants' adaptive behavior.

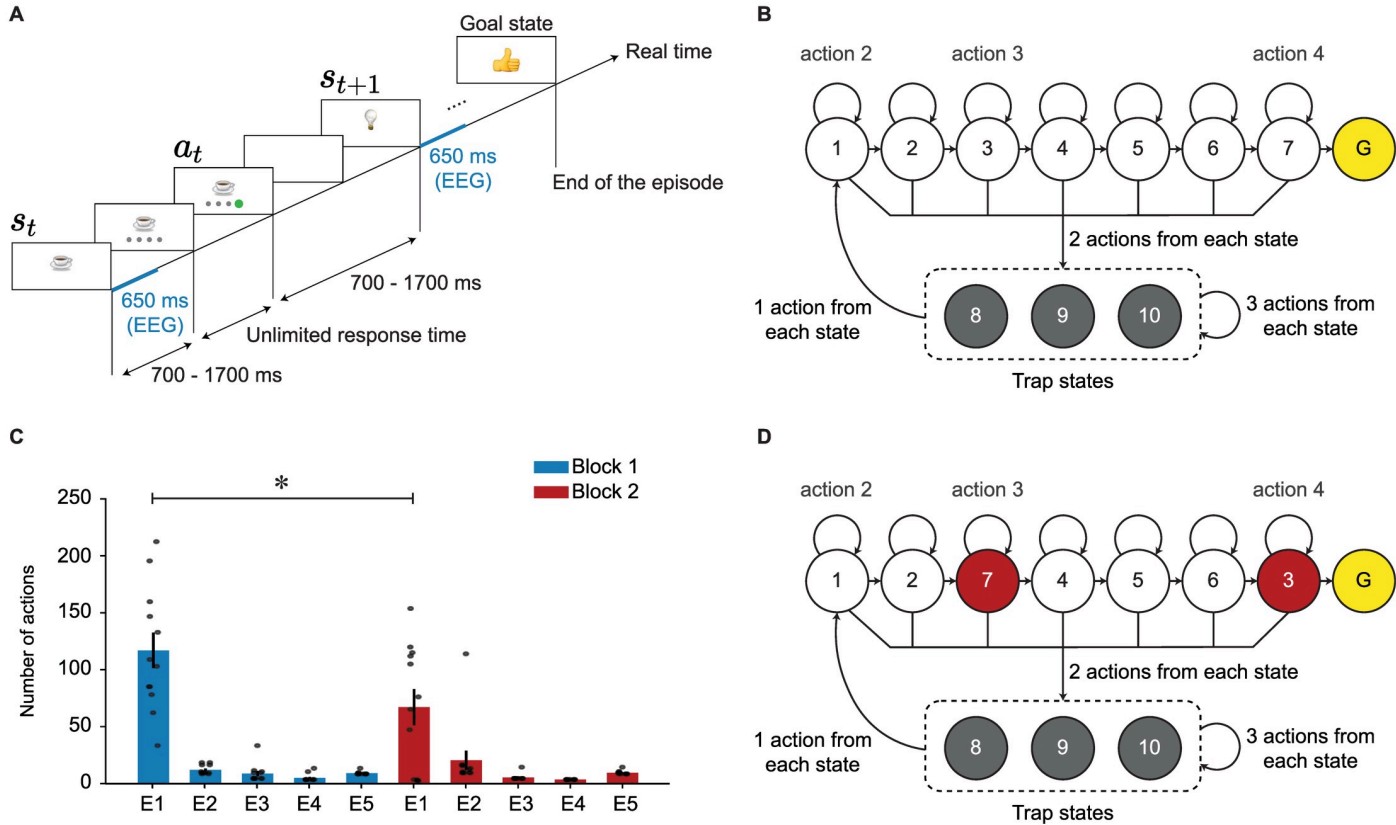

**Fig 1. Experimental paradigm. A**. After image onset, participants had to wait for 700–1700ms (randomly chosen) until four grey disks were presented at the bottom of the image. After clicking on one disk, a blank screen was presented for another random interval of 700 to 1700ms. The next image appeared afterwards. Different participants saw different images, but the underlying structure was identical for all participants. The goal image is a 'thumb-up' image in this example. The blue lines indicate the window of EEG analysis. **B**. Structure of the environment during block 1. There were 10 states with 4 actions each plus a goal state (G). States 1–7 are *progressing states* and states 8–10 are *trap states*. For each progressing state, one action led participants to the next progressing state, two actions led participants to one of the trap states, and one action made participants stay at the current state. The action which made participants stay at the current state is shown for states 1, 3, and 7, as an example. For each trap state, three actions led participants to one of the trap states, and one action led participants to state 1. Not all action arrows are drawn for the trap states to simplify illustration. **C**. Average number of actions of participants during block 1 (blue) and block 2 (red): The 1st episode of block 2 was significantly shorter than the 1st episode of block 1 (one-sample t-test, p-value = 0.035). Error bars show the standard error of the mean, and each grey point shows the data of one participant. **D**. Environment used in block 2: The images presenting state 3 and state 7 (in red) were swapped. Other transitions remained unchanged.

## Results

### Experimental paradigm and human behavior

In order to distinguish between novelty, surprise, and reward, and to study their effects on exploratory and adaptive behavior, we designed an environment (cf. [44]) consisting of 10 states with 4 possible actions per state plus one goal state (Fig 1A and 1B). In the human experiments, states were represented as images on a computer screen and actions as four grey disks below the image. Before the experiment, 12 participants were shown all images of the states and were informed that their task was to find the shortest path to the goal image. Throughout the experiment, at each state, participants chose an action (by clicking on one of the grey disks) which brought them to the next image, where they then chose the next action, and so on (Fig 1A). Such an episode ended when the goal image was found.

Unknown to the participants, the non-goal states could be classified into the progressing states (1 to 7 in Fig 1B) and the trap states (8 to 10 in Fig 1B). At each progressing state, one action ('good' action) either brought participants to another progressing state closer to the

goal or led them directly to the goal, two actions ('bad' actions) brought them to one of the trap states, and one action ('neutral' action) made them stay at the current state. At each trap state, three actions brought participants to either the same or another trap state, and one action brought them to state 1, at the beginning of the path of progressing states. The assignment of action buttons to specific transitions was random and not the same for different states, e.g., in state 1, the neutral action is action 2, whereas in state 3, the neutral action is action 3 (Fig 1B). Note that the underlying structure of the environment, the assignment of images to specific states, and the assignment of action buttons to specific transitions were unknown to the participants. We also did not tell the participants whether or not transitions were deterministic (i.e., whether the same action from a certain state always led to the same next state).

The experiment was organized in 10 episodes, i.e., it ended after the 10th time that participants found the goal state. Unknown to the participants, we divided these 10 episodes into 2 blocks of 5 episodes each; we refer to the first 5 episodes as block 1 and to the second 5 episodes as block 2. During the 1st episode of block 1, participants took between 34 and 214 actions (mean 118 and std 54) until they arrived at the goal (Fig 1C). They then continued for another 4 episodes, each time starting in a new initial state. The initial state for each episode was chosen randomly, but it was kept fixed across participants. After the 1st episode, participants had learnt to reach the goal in less than 20 steps (episodes 2 to 5 in Fig 1C). After the end of the 5th episode (the end of block 1), two states (state 3 and 7 in Fig 1D) were swapped, without announcing it to the participants. Participants continued for another 5 episodes with the novel layout of the environment (2nd block, Fig 1D).

In the 1st episode of block 1, participants explored the environment to find the goal, but they received no intermediate reward or other sign of progress while doing this. If participants followed a purely random exploration (i.e., choosing each action with 1/4 probability), it would take them on average about $10^4$ actions to find the goal, starting at any non-goal state (see S4 Text). This high number is an indication of the complexity and depth of our environment. Our results suggest that participants followed a non-random strategy for finding the goal (Fig 1C). With increasing experience, the latency of escape from the trap states was reduced (Fig 2A) and the good actions at progressing states were chosen with higher probability (Fig 2B). It is important to note that these improvements were observed in the absence of any external feedback indicating progress and before the 1st encounter of the goal state. Here, we ask whether novelty of states played a role in the way participants chose their actions and searched for the goal.

In the 1st episode of block 2, when states 3 and 7 had been swapped, participants spent a great amount of time (68 ± 16 actions on average) re-exploring the environment and searching for the goal state, but they were significantly faster in finding the goal than in the 1st episode of block 1 (Fig 1C). After the swap, participants continued escaping from the trap states (Fig 2A) and choosing the good actions at the unchanged progressing states (states 1, 2, and 4 in Fig 2C). Moreover, they rapidly adapted their behavior and found the new good actions at the swapped states (state 7 in Fig 2C). Our results indicate that participants adapted their behavior to the new situation while exploiting the knowledge they had acquired before. This observation suggests that surprise triggered by unexpected transitions helped participants to rapidly adapt their behavior. Here, we ask how surprise affects participants' adaptive behavior.

## Defining novelty and surprise

In the Oxford English Dictionary [45], novelty is defined as 'the quality or state of being new, original, or unusual'. Here, our focus is on the quality of being 'unusual', and by saying that a state is novel, we mean that it has not been encountered often, i.e., it is not 'usual' to encounter

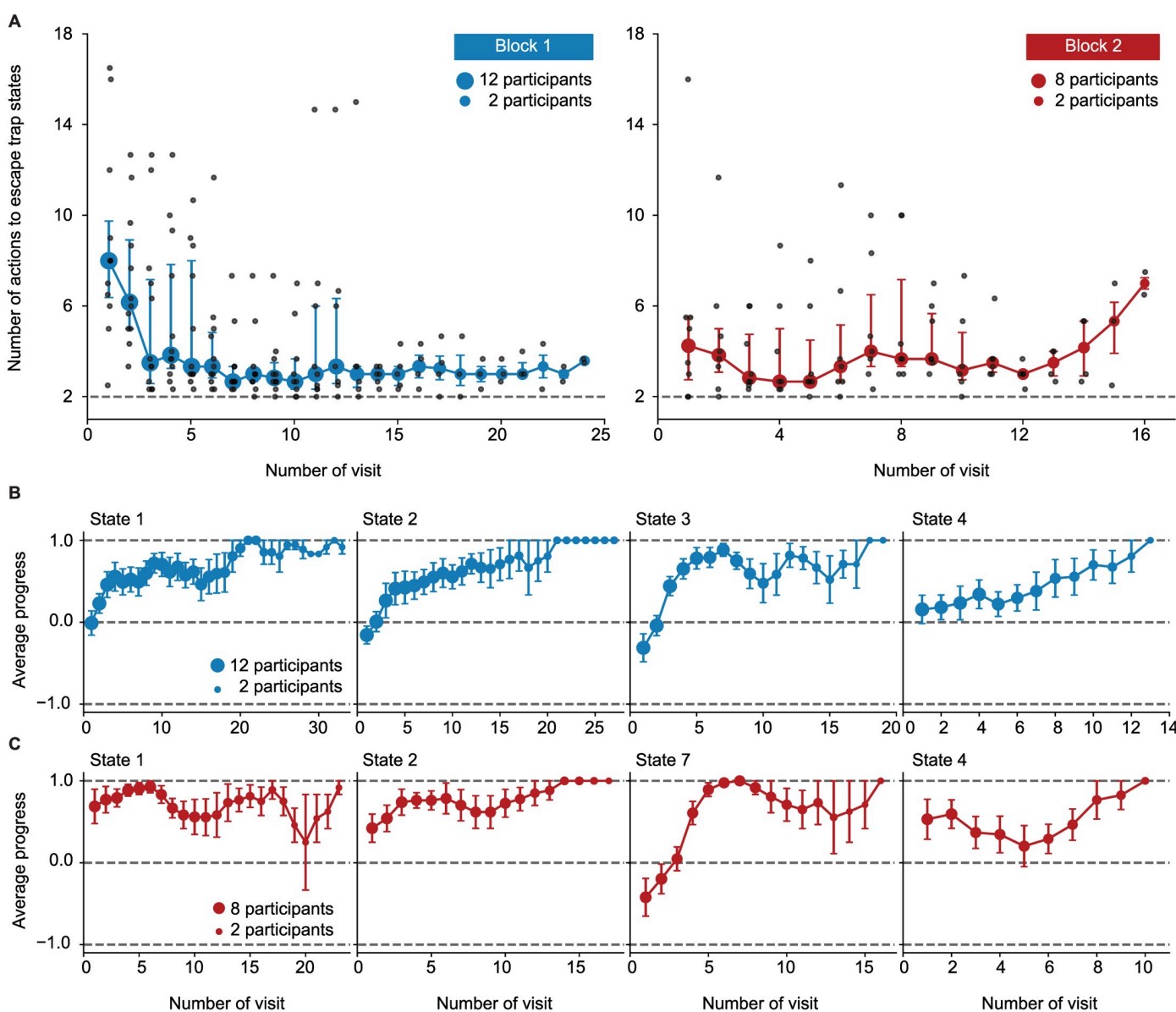

**Fig 2. Behavioral results for episode 1 of blocks 1 and 2. A**. Escape from the trap states: Median number of actions of participants between falling into a trap state and reaching state 2 in episode 1 of block 1 (left) and block 2 (right). Error bars show the 25% and 75% quantiles, and each grey point shows the data of one participant. The grey dashed lines correspond to the minimum number of actions (2) that are needed to escape the trap states. x-axis shows the number of visits of the trap states, for example, 10 means the 10th times participants fall from a progressing state into the trap states. Because of between-participant differences, not all participants visited the trap states for, e.g., 20 times. The size of circles indicates number of participants over which the average is taken. In the 1st episode of block 2 (right), four participants reached the goal state without falling into the trap states; thus, only the data for the other 8 participants is shown. A moving average of length three was applied to the data. **B**. Average progress of participants each time visiting states 1, 2, 3, and 4 in episode 1 of block 1. We assign a progress value of 1 to good actions (the ones taking participants closer to the goal), 0.5 to neutral actions (the ones making participants stay where they are), and -0.75 to bad actions (the ones taking participants to the trap states); with this assignment, average progress vanishes for random exploration. The size of circles shows the number of participants over which the average is taken, and error bars show the standard error of the mean. A moving average of length three was applied to the data. **C**. Average progress of participants each time visiting states 1, 2, 7 (swapped with 3), and 4 in episode 1 of block 2. See S1 Fig (A) for the average progress at the progressing states in the proximity of the goal.

this state. We, therefore, assume that (i) the novelty of a state $s$ at time $t$ is a decreasing function of the number $C_s^{(t)}$ of encounters of state $s$ until time $t$, e.g., a state that has been encountered 5 times is less novel than a state that has been encountered only once. Moreover, we assume that (ii) a state $s$ that has been encountered for example $C_s^{(t)} = 5$ times within a total of $t = 5$ trials is

less novel compared to a state $s'$ that has been encountered for example $C_{s'}^{(t')} = 5$ times within a total of $t' = 500$ trials. Following assumptions (i) and (ii), we define the novelty of a state $s$ as a decreasing function of the observation frequency

$$p_N^{(t)}(s) = \frac{C_s^{(t)} + 1}{\left(\sum_{s'} C_{s'}^{(t)}\right) + 11}. \tag{1}$$

$p_N^{(t)}(s)$ has two different interpretations. First, it can be seen as the empirical frequency of observing state $s_t$ until time $t$. In fact, because one of the counters $C_{s'}$ increases by one at each time step, the time can be expressed as $t = \sum_{s'} C_{s'}^{(t)}$. In this interpretation, the numbers 1 in the numerator and 11 (11 is the total number of states in the environment) in the denominator correspond to the one encounter of each state before the start of the experiment. In the second interpretation, $p_N^{(t)}(s)$ can be seen as the probability of observing state $s$ at time $t$, estimated in a Bayesian framework and with the assumption of independence between observations (see S1 Text); measures similar to $p_N^{(t)}(s)$, sometimes called 'density models', have been used in machine learning, for example, to quantify how frequently an image has been observed [17]. In the Bayesian interpretation, the numbers 1 in the numerator and 11 in the denominator correspond to a uniform prior that makes all states equally likely at time $t = 0$.

We define the novelty of state $s$ at time $t$ as

$$N^{(t)}(s) = -\log p_N^{(t)}(s). \tag{2}$$

Consistent with the literature [35, 46, 47], we chose the logarithm in order to smooth out temporal fluctuations and compress differences in the novelty of frequent versus infrequent states (Fig 3A). Since our novelty measure depends on the frequencies (relative counts, $p_N^{(t)}(s)$ in Eq 1) rather than the raw counts ($C_s^{(t)}$), one may also interpret $N^{(t)}(s)$ as a measure of 'relative novelty' or 'rareness'. See Discussion for the relation of our measure of novelty to other measures.

With our definition of novelty, at the beginning of the 1st episode in block 1, all states have identical novelty. Since participants often fall into one of the trap states, the novelty of the trap states decreases rapidly (Fig 3A). Hence, before the end of the 1st episode, the novelty is highest for states in the proximity of the goal (Fig 3B). This observation suggests that seeking novel states will, in our environment, effectively lead a participant closer to the goal, even *before* the participant knows where the goal is located, i.e., before encountering the goal for the first time. We conclude that novelty is potentially an important signal and will exploit this insight further below.

Surprise is defined in the Oxford English Dictionary [48] as 'the feeling or emotion excited by something unexpected' or 'the feeling or mental state, akin to astonishment and wonder, caused by an unexpected occurrence or circumstance'. Whereas novelty is about being unusual, surprise is about being unexpected. Following this intuition, we define surprise as a measure expressing how 'unexpected' the next image (state $s_{t+1}$) is given the previous state $s_t$ and the chosen action $a_t$. To quantify expectations, we assume that participants build an internal model of the environment ('world-model'), i.e., we hypothesize that participants estimate the probability $p^{(t)}(s_{t+1}|s_t, a_t)$ of a transition from a given state $s_t$ to another state $s_{t+1}$ when performing action $a_t$. More precisely, we assume that the world-model counts transitions from state $s$ to $s'$ under action $a$ using either a leaky [23, 49, 50] or a surprise-modulated [28, 29, 51] counting procedure, described by the pseudo-count $\tilde{C}_{s,a \to s'}^{(t)}$. The conditional probability is then

$$p^{(t)}(s_{t+1}|s_t, a_t) = \frac{\tilde{C}_{s_t, a_t \to s_{t+1}}^{(t)} + \epsilon}{\tilde{C}_{s_t, a_t}^{(t)} + 11\epsilon}, \tag{3}$$

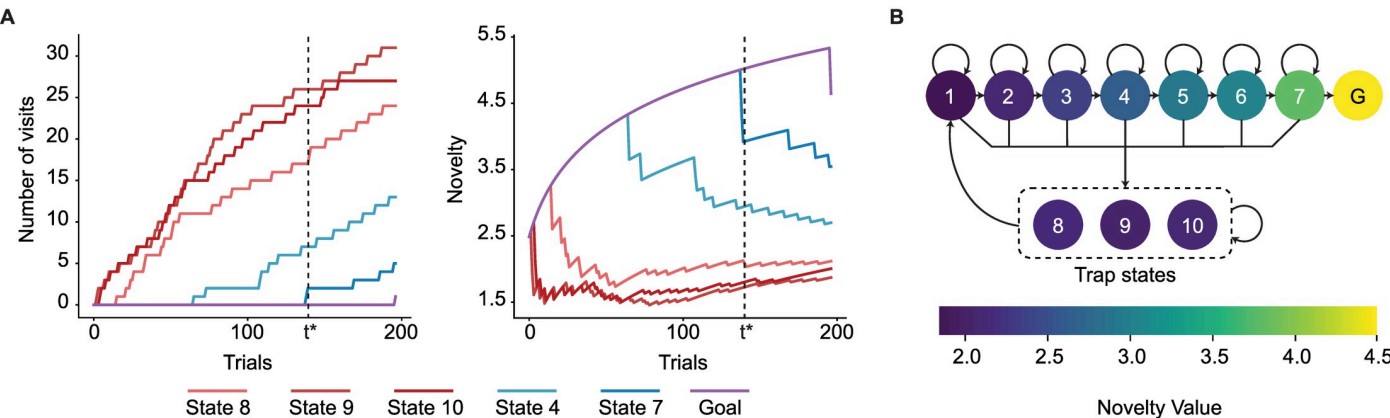

**Fig 3. Novelty in episode 1 of block 1. A**. The number of state visits (left panel) and novelty (right panel) as a function of time for one representative participant: The number of visits increases rapidly for the trap states and remains 0 for a long time for the states closer to the goal. Novelty of each state is defined as the negative log-probability of observing that state (see Eqs 1 and 2) and, hence, increases for states which are not observed as time passes. The first time participants encounter state 7 (the state before the goal state) is denoted by $t^*$. **B**. Average (over participants) novelty (color coded) at $t^*$: Novelty of each state is a decreasing function of its distance from the goal state.

where $\epsilon$ is a parameter corresponding to a prior in the Bayesian framework, 11 is the total number of states in the environment, and $\tilde{C}_{s_t,a_t}^{(t)} = \sum_{s'} \tilde{C}_{s_t,a_t \to s'}$ is the pseudo-count of taking action $a_t$ at state $s_t$ (see Methods and S1 Text). If there is no linear or nonlinear filtering (e.g., leaky integration) applied during the counting process, pseudo-counts are equal to real counts.

Higher values of the conditional probability $p^{(t)}(s_{t+1}|s_t, a_t)$ indicate that a participant expects to experience the transition from the pair of state and action $(s_t, a_t)$ to the next state $s_{t+1}$ with higher probabilities and, hence perceives this transition as less surprising. Therefore, we consider the surprise of such a transition to be a decreasing function of $p^{(t)}(s_{t+1}|s_t, a_t)$. More precisely, we use a recent measure of surprise motivated by a Bayesian framework for learning in volatile environments, called the 'Bayes Factor' surprise [29]. The Bayes Factor surprise of the transition from state $s_t$ to state $s_{t+1}$ after taking action $a_t$ is

$$\mathbf{S}_{\mathrm{BF}}^{(t+1)} = \frac{\mathrm{const.}}{p^{(t)}(s_{t+1}|s_t, a_t)}, \tag{4}$$

where $p^{(t)}(s_{t+1}|s_t, a_t)$ is the conditional probability of observing state $s_{t+1}$ at time $t + 1$ derived from the present world-model. Our surprise measure is an increasing function of the state prediction error [5] and Shannon surprise [46, 50] (see Methods) and takes high values during the 1st episode of block 2 whenever participants encounter states 3 or 7 or transit from state 3 or 7 to another state (Fig 4A and 4B). See Discussion for the relation of our measure of surprise to other measures.

## The SurNoR algorithm: Distinct contributions of novelty and surprise to behavior

We hypothesize that participants use novelty to explore the environment and surprise to modulate the rate of learning. The hypothesis is formalized in the form of the Surprise-Novelty-Reward (SurNoR) algorithm and tested given the behavioral data of 12 participants.

Novelty in SurNoR plays a role analogous to that of reward. For example, in standard Temporal-Difference (TD) Learning, a reward-based Q-value $Q_R(s, a)$ is associated with each state-action pair $(s, a)$ [19]; the Q-value $Q_R(s, a)$ estimates the mean discounted reward that can be collected under the current policy when starting from state $s$ and action $a$, and the reward

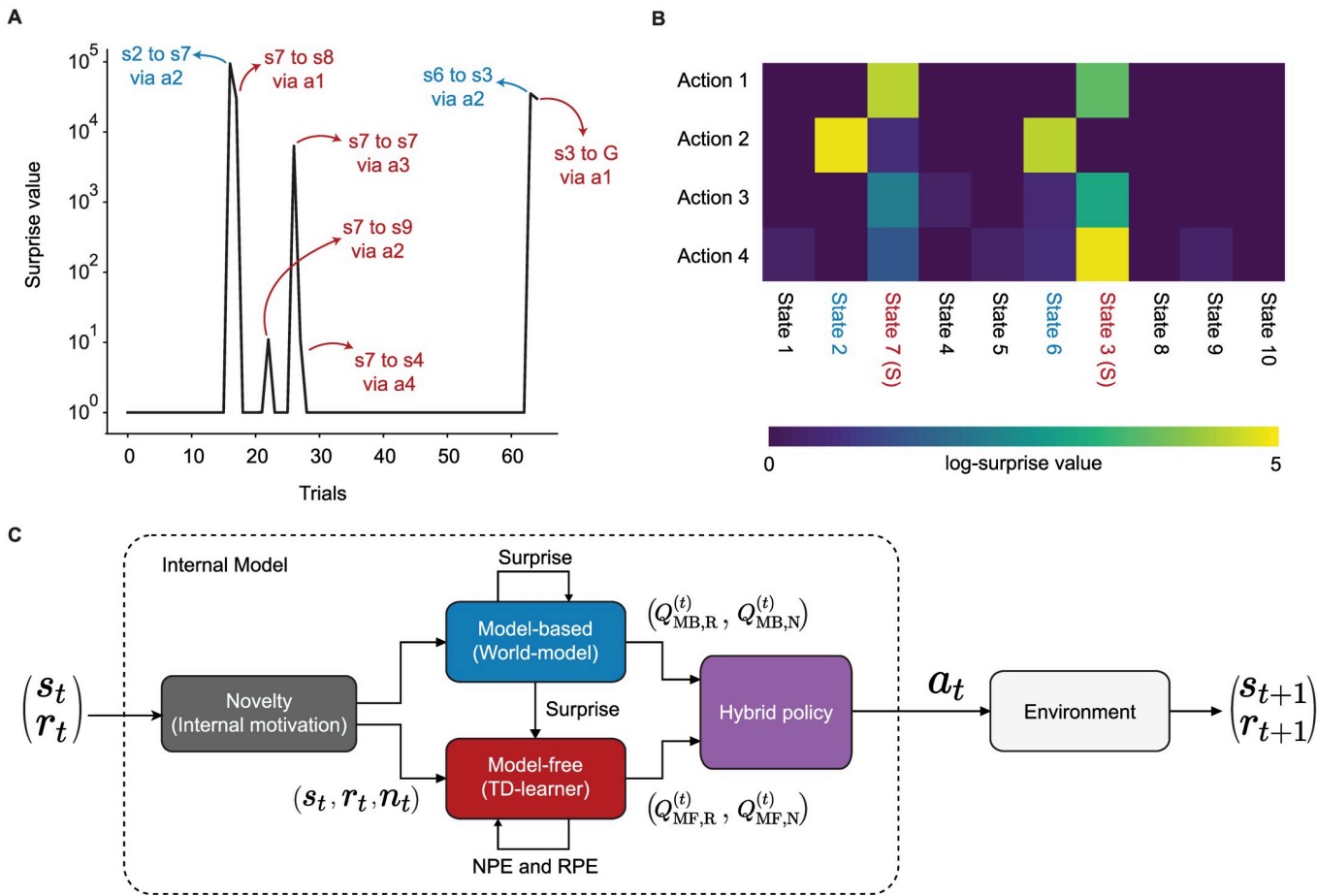

**Fig 4. Surprise as a modulator of the learning rate in episode 1 of block 2. A**. Surprise as a function of time since the start of block 2 for one representative participant: Surprise has very small values most of the time, because the participant has already learned the transitions in the environment during block 1. The surprising transitions are the ones to the swapped states (blue) and the ones from the swapped states (red). **B**. Maximal log-surprise values (yellow = large surprise) during the 1st episode of block 2, averaged over all participants. The swapped states are marked in red and the states before them in blue. One action from each swapped state is not surprising, i.e., the action leading participants to trap states both before and after the swap. **C**. Block diagram of the SurNoR algorithm: Information of state $s_t$ and reward $r_t$ at time $t$ is combined with novelty $n_t$ (grey block) and passed on to the world-model (blue block, implementing the model-based branch of SurNoR) and TD learner (red block, implementing the model-free branch). The surprise value computed by the world-model modulates the learning rate of both the TD-learner and the world-model. The output of each block is a pair of Q-values, i.e, Q-values for estimated reward $Q_{MF,R}$ and $Q_{MB,R}$ as well as for estimated novelty $Q_{MF,N}$ and $Q_{MB,N}$. The hybrid policy (in purple) combines these values.

prediction error RPE, derived from $Q_R(s, a)$, serves as a learning signal even for states a few steps away from the goal [19]. Analogously, in the SurNoR model, novelty is a reward-like signal with associated novelty-based Q-values $Q_N(s, a)$ and an associated novelty prediction error (NPE) derived from $Q_N(s, a)$. In the SurNoR model, the two sets of Q-values, reward-based and novelty-based, are used in a hybrid model [5, 6] that flexibly combines model-based with model-free action selection policies (Fig 4C).

Surprise in SurNoR is derived from a mismatch between observations of the next state and predictions arising from the world-model embedded in the model-based branch of SurNoR. To adapt both model-based and model-free policies of the SurNoR algorithm, surprise is used in two different ways. First, high values of surprise systematically lead to a larger learning rate for the update of the world-model than smaller ones, consistent with earlier models [27, 29]. Second, going beyond previous models of behavior [20, 24–26, 30], surprise also influences the learning rate of the model-free reinforcement learning branch.

We predict that, if the behavior of participants is well described by the SurNoR algorithm, they should use an action policy that attracts them to novel states, in particular during the 1st episode of block 1. If participants do not exploit novelty, standard (potentially hybrid) reinforcement learning schemes in combination with one of several alternative exploration strategies (see next section) should be sufficient to explain the behavior. Furthermore, we predict that, if the behavior of participants is well described by the SurNoR algorithm, then surprising events during the 1st episode of block 2 should significantly change the behavior of participants; if participants do not exploit surprise, standard hybrid models combining model-based and model-free reinforcement learning [5, 6] should be sufficient to describe the behavior.

## Both surprise and novelty are needed to explain behavior

SurNoR has three main components: (i) action selection by hybrid policy, (ii) exploration by novelty-seeking, and (iii) learning by surprise-modulation. To test our hypothesis, and to test whether all three components of SurNoR are necessary for explaining behavior or whether a simpler or an alternative model would have the same explanatory power, we compared SurNoR with 11 alternative algorithms plus a null algorithm based on a random choice (RC) of actions (Fig 5A). Three out of 11 algorithms use a hybrid policy (+Hyb), five use novelty-seeking (+N), and seven use surprise-modulation (+S).

Alternatives for (i) action selection were pure model-based (MB; 4 out of 11 algorithms) and pure model-free (MF; 4 out of 11) policies. Note that we allow for the possibility that MF algorithms are equipped with a world-model for computation of surprise but do not use this world-model for action-planning. As alternatives for (ii) exploration strategy, we used optimistic initialization (+OI; 3 out of 11) [19] and uncertainty (surprise) seeking [42, 43] (+U; 3 out of 11); see below for more explanations and S1 Text for details. Finally, as an alternative to surprise modulation, we used constant learning rates for learning the world-model and model-free Q-values [22, 23, 49, 50] (all algorithms without +S; 4 out of 11). For the details of the alternative algorithms see S1 Text.

Given the behavioral data of all 12 participants, we estimated the log-evidence of all 13 algorithms, including SurNoR (see Methods). Comparison of the algorithms' log-evidence (Fig 5A) shows that SurNoR explains human behavior significantly better than its alternatives. In addition, a Bayesian model selection approach with random effects [52, 53] indicates that the SurNoR algorithm outperforms the alternatives with a protected exceedance probability of 0.99 (Fig 5B and Methods).

The 1st episode of the 1st block is ideally suited to study how novelty influences behavior (middle panel in Fig 5A). Our results show that all algorithms with novelty-seeking (+N) explain the behavior significantly better than models with random exploration strategy (RC) or optimistic initialization (MB+S+OI, MF+OI, and Hyb+S+OI), i.e., two classic approaches for exploration [19]. Our results also show that novelty-seeking explains behavior better than uncertainty-seeking (+U), a state-of-the-art exploration method in reinforcement learning [42, 43]. The models with uncertainty-seeking (MB+S+U, MF+S+U, and Hyb+S+U) use surprisal (i.e., the logarithm of our surprise measure) as an intrinsic reward as opposed to our model of novelty-seeking that uses novelty of states as an intrinsic reward.

As an alternative to novelty-seeking, participants might also solve the task simply by detecting and avoiding trap states. If so, the behavior of the participants can be explained if we replace the continuous novelty signal by a simple intrinsically generated binary signal equivalent to a negative reward. To address this issue, we tested two modified versions of the SurNoR algorithm ('Binary Novelty', see S1 Text). The 1st modification detects those states that have been encountered more often than some threshold value and assigns a fixed negative reward

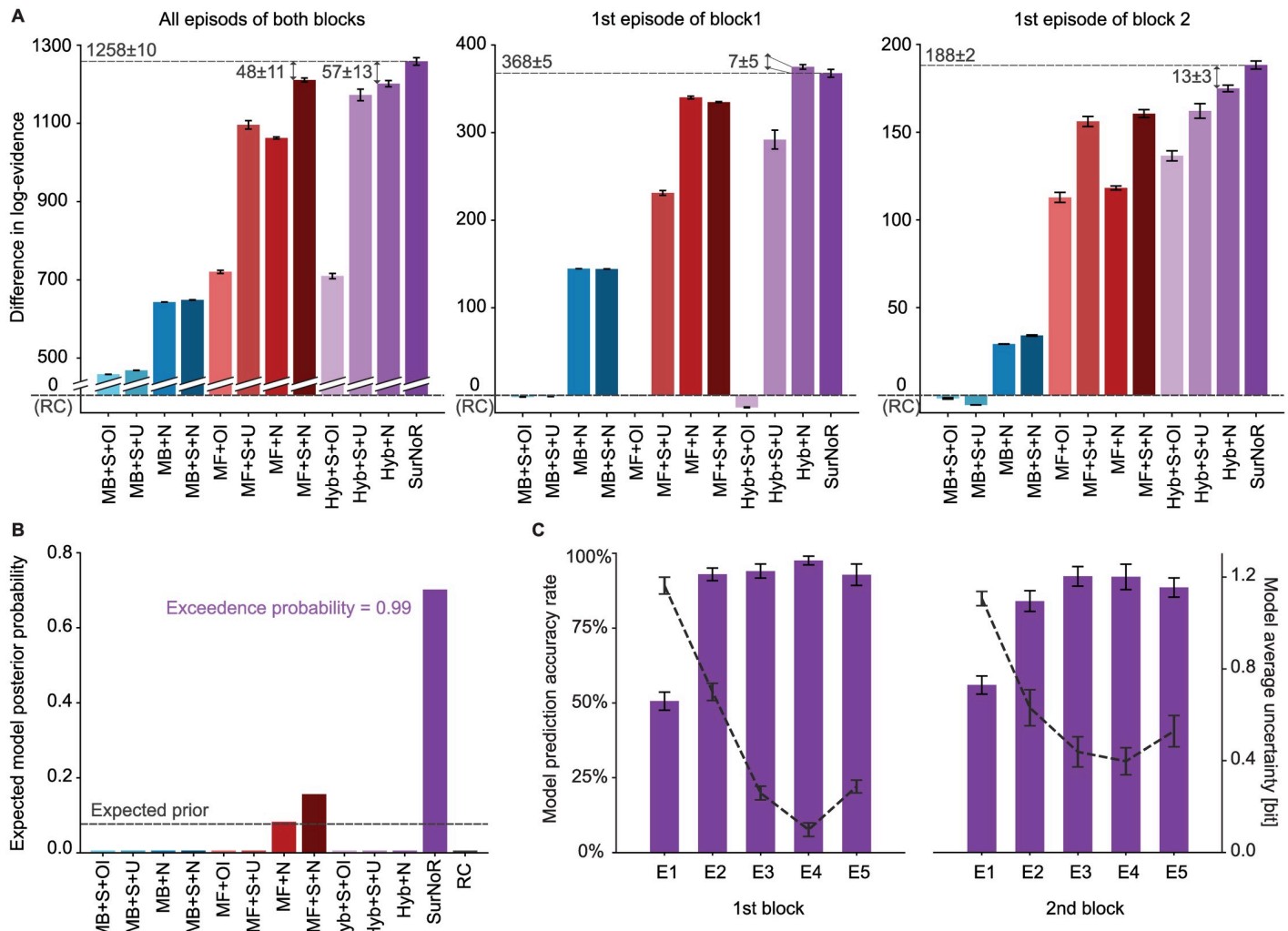

**Fig 5. Model comparison of model-based (MB, blue bars), model-free (MF, red bars), and hybrid algorithms (Hyb and SurNoR, purple bars).** Exploratory behavior is either induced by optimistic initialization (+OI), uncertainty-seeking (+U), unbiased random action choices (RC), or novelty-seeking (+N); e.g., a model-based algorithm with novelty seeking is denoted as MB+N. SurNoR and the model-free or hybrid algorithms annotated with '+S' use surprise to modulate the learning rate of the model-free TD learner; SurNoR and all algorithms annotated with '+S' use surprise modulation also during model building (see Methods). **A**. Difference in log-evidence (with respect to RC) for the algorithms for all episodes of both blocks (left panel), the 1st episode of block 1 (middle), and the 1st episode of block 2 (right panel). High values indicate good performance; differences greater than 3 or 10 are considered as significant or strongly significant, respectively (see Methods); a value of 0 corresponds to random action choices (RC). The random initialization of the parameter optimization procedure introduces a source of noise, and the small error bars indicate the standard error of the mean over different runs of optimization (Methods, statistical model analysis). **B**. The expected posterior model probability [52, 53] given the whole dataset (Methods) with random effects assumption on the models. **C**. Accuracy rate of actions predicted by SurNoR (left scale and purple bars: mean and the standard error of the mean across participant) and the average uncertainty of SurNoR (right scale and dashed grey curve: mean entropy of action choice probabilities and the standard error of the mean across participants).

to them. The 2nd modification considers the *n* most frequently encountered states as bad states and, similar to the 1st modification, assigns a fixed negative reward to them—where *n* is a free parameter of the algorithm. Note that in both control algorithms, the constant negative rewards are treated as an intrinsic motivation signal—similar to novelty in SurNoR-algorithm except that the signal is a binary one. We estimated the log-evidence for both control algorithms. Our results show that SurNoR outperforms the 1st control algorithm by a 244 ± 11 difference in total log-evidence and by a 235 ± 5 difference in the log-evidence of the 1st episode of block 1, and outperforms the 2nd control algorithm by a 240 ± 11 difference in total

log-evidence and by a 234 ± 5 difference in the log-evidence of the 1st episode of block 1. This observation rejects the hypothesis that participants simply identify 'bad' states by some binary signal.

Surprise becomes important in the 1st episode of block 2 (right panel in Fig 5A). Indeed, the SurNoR model is significantly better than a hybrid model with novelty but without surprise (Hyb+N); similarly, model-free reinforcement learning with novelty and surprise (MF+S+N) is significantly better than model-free reinforcement learning with novelty alone (MF+N, right panel in Fig 5A). Our results show that a constant adaptation rate as implemented in standard models without surprise is not sufficient to explain the choices of participants in the episode after the swap. Rather, the rate of learning and forgetting has to be modulated by a measure of surprise.

Overall, SurNoR is better than all 12 competing algorithms by a large margin, indicating that a combination of model-based and model-free algorithms explains behavior better than each algorithm separately, consistent with the notion of parallel, model-based and model-free, policy networks in the brain [3, 5, 6]. Going beyond these earlier studies, our results with SurNoR indicate that surprise and novelty are both necessary to explain human behavior in our task. Novelty is necessary to explain behavior during phases of exploration while surprise is necessary to explain behavior during the rapid re-adaption after a change in the environment.

## Individual decisions are dominated by the model-free policy network

We wondered whether the SurNoR model is also able to predict the individual actions of participants. Taking the most probable action of the model in a given state as the prediction of a participant's next action in that state, SurNoR predicted the correct action in the 1st episode of block 1 with an accuracy of 51 ± 3% (3-fold cross validated, mean ± standard error of the mean over 12 participants, see Methods—Fig 5C). Note that this accuracy is achieved in the absence of any *a priori* preference of actions at initialization and is significantly higher than the accuracy rate of the naive random exploration strategy (25%, chance level).

SurNoR's predictions are also significantly better than the predictions of directed exploration through optimistic initialization (OI) or an uncertainty-seeking policy (U). Models with OI could at best predict 36 ± 3% of the actions (for MB+S+OI), and the uncertainty-seeking strategy could at best predict 46 ± 3% (for Hyb+S+U); one-sample t-test p-values for comparing their accuracy rates versus SurNoR's are 0.01 and 0.0025, respectively. A crucial difference between OI and novelty-based exploration is that OI prefers those actions that have been less frequently chosen in the past, while novelty-seeking prefers actions that lead to novel states, even if these are a few actions ahead and the outcome of the current action is known. Uncertainty-seeking is similar to OI because the uncertain actions are also those that have been less frequently chosen in the past.

Similarly, in the 1st episode of block 2, after the swap of states 3 and 7, the SurNoR algorithm predicts 56 ± 3% of the actions of the 12 participants (Fig 5C). In the remaining episodes 2–5 of the two blocks, the SurNoR algorithm predicts 89 ± 2% of the action choices (Fig 5C). Most of these actions move participants closer to the goal. The intrinsic uncertainty of action choices with the SurNoR model can be estimated from the entropy of the action choice probabilities across the four possible actions (Fig 5C). Uncertainty decreases during the first three episodes as participants become familiar with the environment, but it jumps back to higher values after the swap of states at the beginning of block 2.

To see what aspects of behavior the different components (i.e., hybrid policy, surprise, and novelty) of SurNoR capture, we fitted the parameters of the SurNoR algorithm to the behavior of the 12 participants (see Methods). Since SurNoR combines in its hybrid action selection

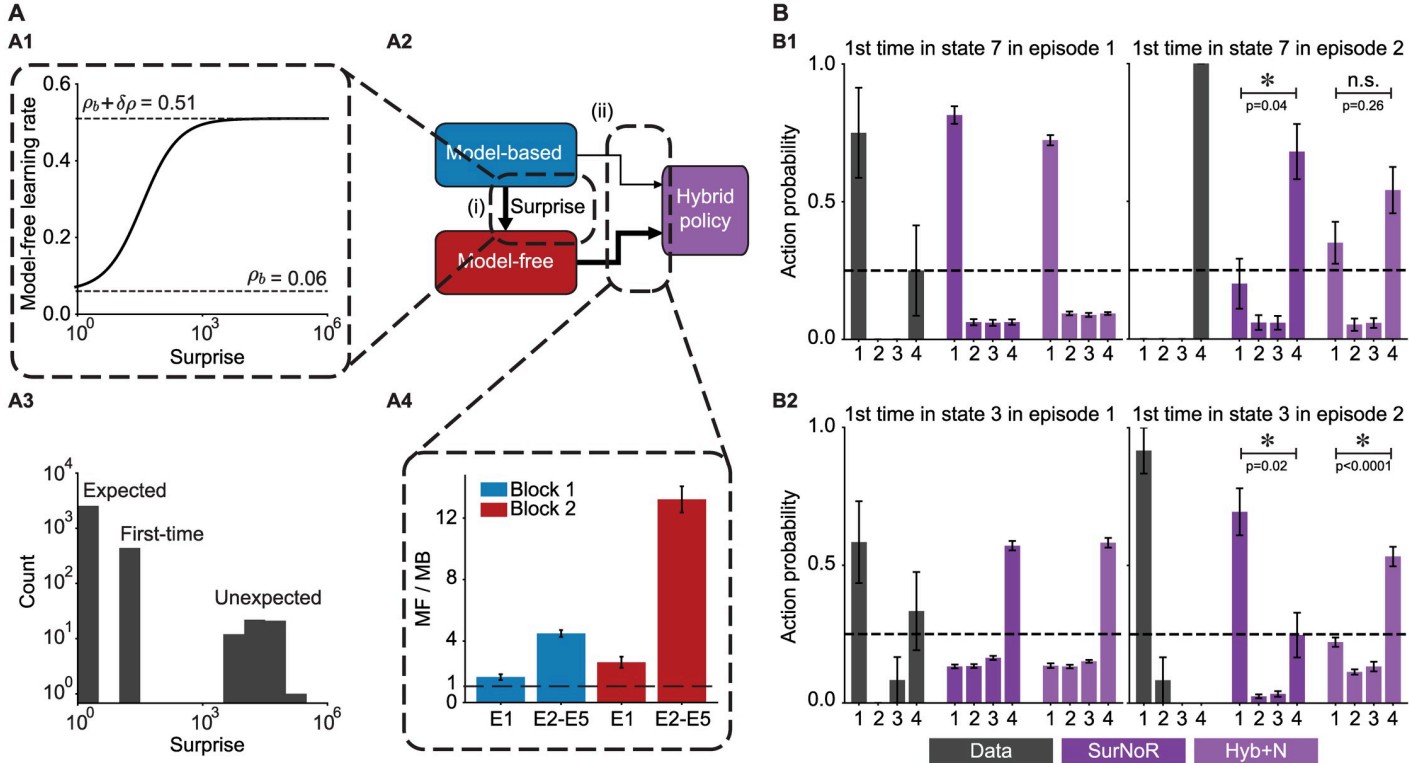

**Fig 6. A. Model-based surprise modulates model-free learning. A1.** The learning rate of the model-free branch as a function of the model-based surprise, after fitting parameters to the behavior of all participants (see Eq 9 in Methods). The model-free learning rate for highly surprising transitions is more than 8 times greater than the one for expected transitions. **A2.** Three modules from the block diagram of Fig 4C. There are two types of interactions between the model-based and the model-free branches of SurNoR: (i) The model-based branch modulates the learning rate of the model-free branch and (ii) the weighted (arrow thickness) outputs of the model-based and the model-free branches influence action selection (hybrid policy). **A3.** The histogram of surprise values across all trials of 12 participants. The distribution is multimodal with high surprise for the unexpected transitions in the 1st episode of block 2, medium surprise for whenever a transition is experienced for the first time, and low surprise for the expected transitions. **A4.** The relative importance of model-free (MF) compared to model-based (MB) in the weighting scheme of the hybrid policy during different episodes. Vertical axis: dominance of model-free (see Methods). Values larger than one (dashed line) indicate that the model-free branch dominates action selection. Error bars show the standard error of the mean. **B. Action choice probability indicates that surprise boosts learning during a single episode**. Action choice probabilities of participants (data, grey) are compared with those of SurNoR and Hyb+N at the first time visiting state 7 (**B1**) or state 3 (**B2**) in episodes 1 (left) and 2 (right) of block 2. **B1.** In state 7, action 1 is the good action before the swap, and action 4 is the good action after the swap. Error bars show the standard error of the mean, and the black dashed line corresponds to random choice action probability (0.25). In episode 2, SurNoR assigns a significantly higher probability to action 4 than to action 1, while according to the Hybrid model without surprise modulation, the action probabilities of action 1 and action 4 are not significantly different. **B2.** In state 3, action 4 is the good action before the swap, and action 1 is the good action after the swap. Behavioral data and SurNoR show a more rapid re-adaptation to the good action than Hyb+N. Note that only 8 (out of the 12) participants encountered state 7 in the first episode of block 2 before reaching the goal. We therefore limit the data analysis to these 8 participants in B1 but use data of all 12 participants in B2.

policy a model-free with a model-based component (Fig 6A2), we first wanted to analyze the relative importance of each of the two components in explaining the action choices of participants; see S6 Text for a qualitative comparison of these two components. In order to evaluate the relative importance of the two components, we normalized the Q-values of both branches and determined the relative weight of each branch (see Methods) during the 1st episode and 2nd-5th episodes of each block (Fig 6A4). We find that the model-free branch dominates the actions. Thus the world-model is of secondary importance for action selection and is mainly used to detect surprising events.

Second, in order to quantify the influence of surprise on learning, we plot the learning rate (of the model-free Q-values $Q_{MF,R}$ and $Q_{MF,N}$) as a function of surprise (Fig 6A1). We find that non-surprising events lead to a small learning rate of 0.06 whereas highly surprising events induce a learning rate that is more than 8 times higher (Fig 6A1 and 6A3) indicating that

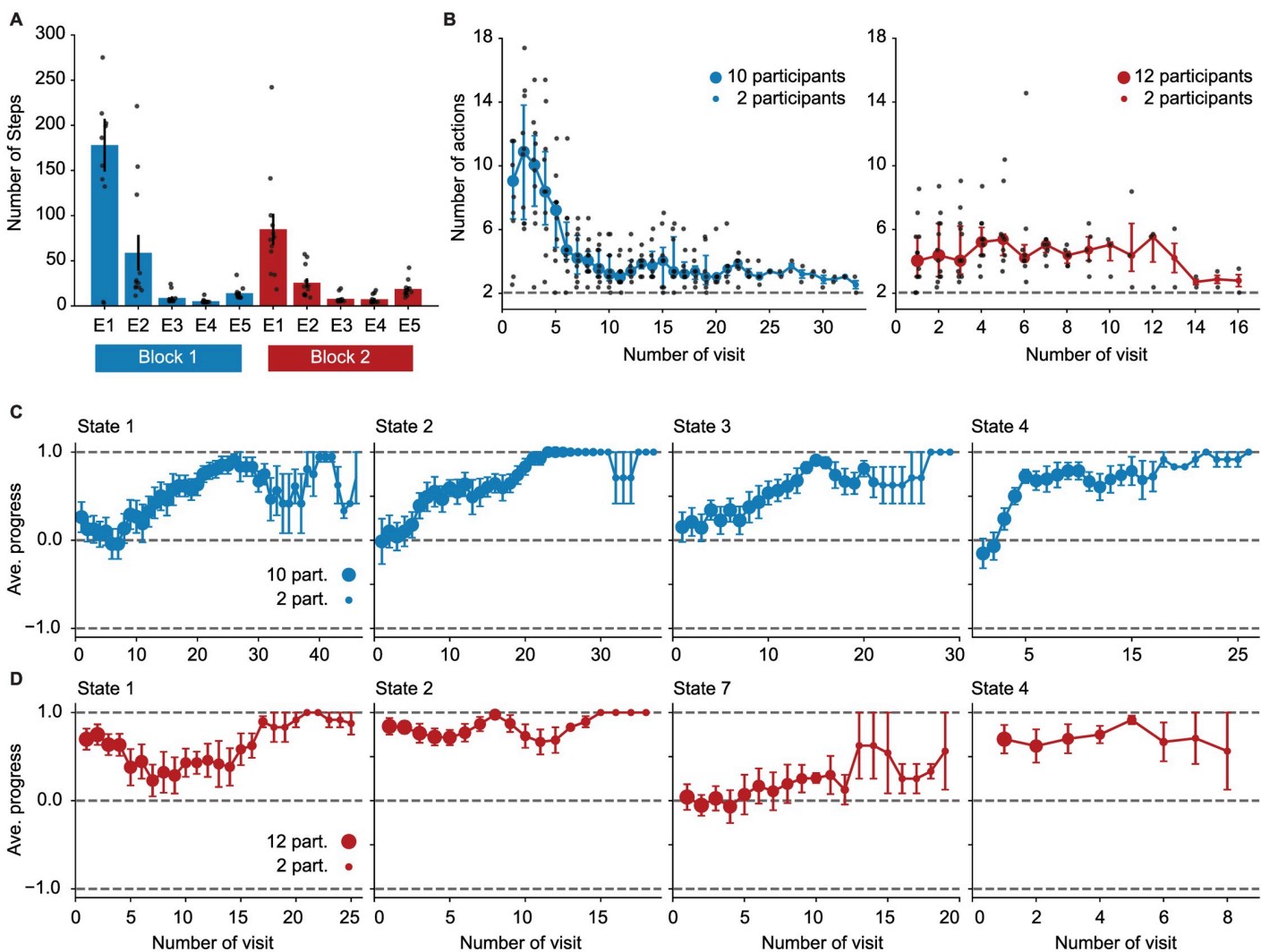

**Fig 7. Posterior predictive checks. A**. Average number of actions of all 12 simulated participants for each episode (c.f. Fig 1C). **B**. Median number of actions of simulated participants to escape the trap states at each of their visits in episode 1 of block 1 (left) and block 2 (right) (c.f. Fig 2A) **C**. Average progress of participants each time visiting states 1, 2, 3, and 4 in episode 1 of block 1. (c.f. Fig 2B). **D**. Average progress of simulated participants each time visiting states 1, 2, 7 (swapped with 3), and 4 in episode 1 of block 2. (c.f. Fig 2C). See S2 and S3 Figs for two other sets of 12 simulated participants with different random seeds. See S1 Fig (B) for the average progress at the progressing states in the proximity of the goal.

surprise strongly influences the update of model-free Q-values. Moreover, we compared the action choices of participants with those of SurNoR and the model Hyb+N (i.e., SurNoR without surprise-modulation) at the swapped states in the 1st and 2nd episode of block 2 (Fig 6B). Our results show that the modulation of the learning rate by surprise in SurNoR is necessary to explain the rapid adaptation of participants after the switch of states.

Finally, to see if the SurNoR model captures, in addition to other aspects, also the exploratory behavior of participants (Figs 1C and 2), we computed posterior predictive checks [54, 55]. To do so, we simulated SurNoR with its parameters fitted to behavior and generated data for 12 simulated participants; see Fig 7 for one set of 12 simulated participants and S2 and S3 Figs for two other sets with different random seeds. Our results show that several important features of the behavior of participants are observed also in the behavior of the simulated participants: (i) They are faster in finding the goal in the first episode of block 2 than in the first

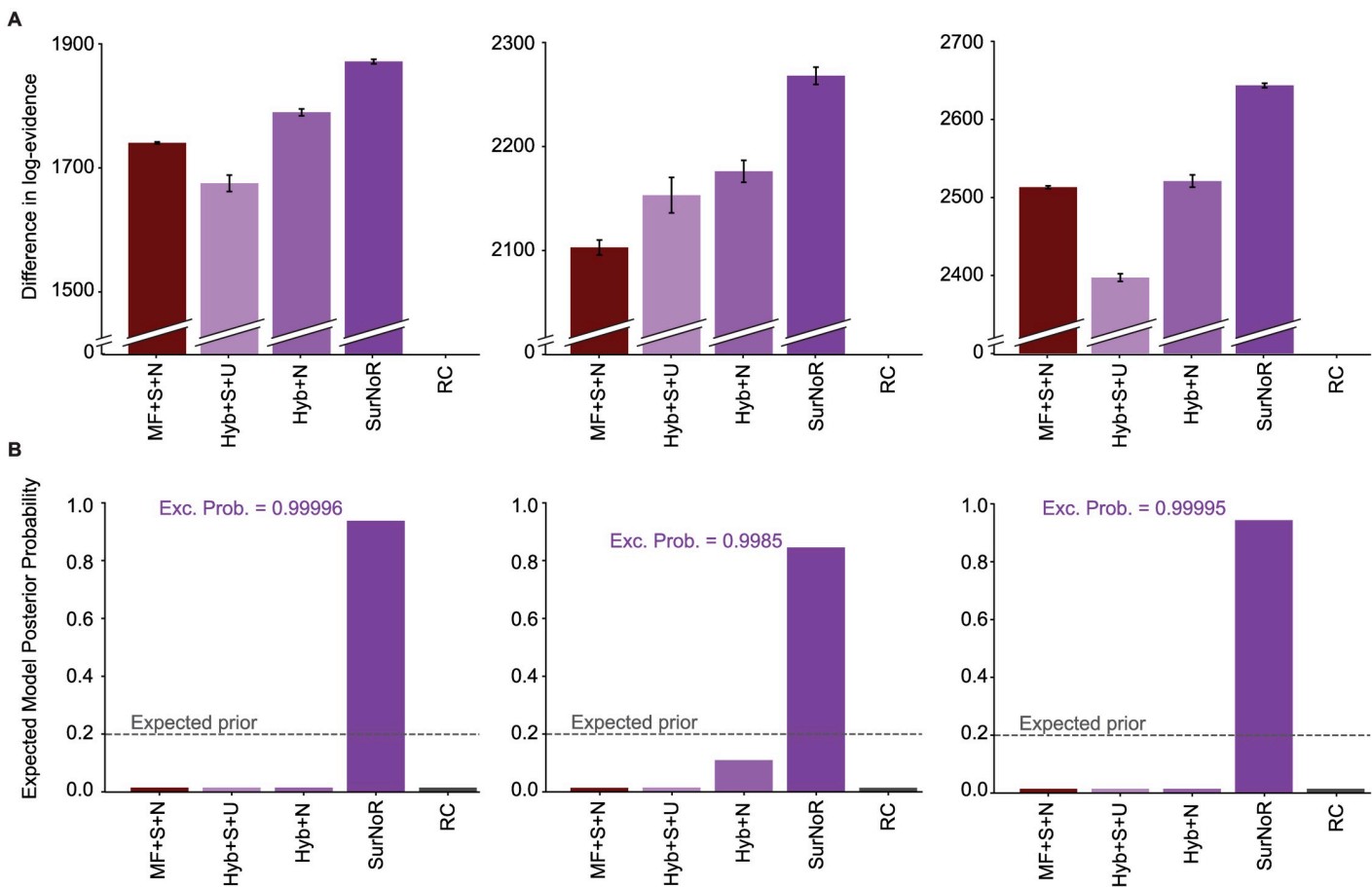

**Fig 8. When data is generated by SurNoR, the true model can be recovered.** We applied our model-selection method to the data of three sets of 12 simulated participants. The left column corresponds to the data shown in Fig 7, and the middle and the right columns correspond to the data shown in S2 and S3 Figs, respectively. We compared the SurNoR model with the strongest competitors of SurNoR: MF+S+N, Hyb+S+U, and Hyb+N (c.f. Fig 5). **A.** Difference in log-evidence with respect to random choice (RC) and **B.** the expected posterior model probability [52, 53] for the algorithms for all episodes of both blocks given the data of each of the three sets (different columns) of 12 simulated participants (c.f. Fig 5A and 5B).

episode of block 1 (Fig 7A), (ii) they learn to escape the trap states and to choose the good action at progressing states in the 1st episode of block 1 (Fig 7B and 7C), and (iii) after the swap, they continue choosing the same actions at unchanged states but rapidly unlearn previously learned actions at the swapped states (Fig 7B and 7D). Moreover, our model-selection approach can successfully recover the true model (SurNoR) given the data of 12 simulated participants (Fig 8). This observation shows that our experimental paradigm is capable of differentiating between SurNoR and its alternatives, and as few as 12 participants are sufficient for drawing conclusions based on our model-selection results [55] (Fig 5); see S3 Text and S4 and S5 Figs for parameter recovery analysis.

In conclusion, the SurNoR algorithm is able to capture different aspects of participants' behavior and to predict individual actions with a high accuracy: it predicts 63 ± 2% of all actions and 74 ± 3% of the actions after the first time finding the goal. Our results suggests that participants (i) rely on propagation of novelty information via NPE in the first episode, (ii) base their decisions mainly on the model-free learner, and (iii) use surprise to modulate the learning rate.

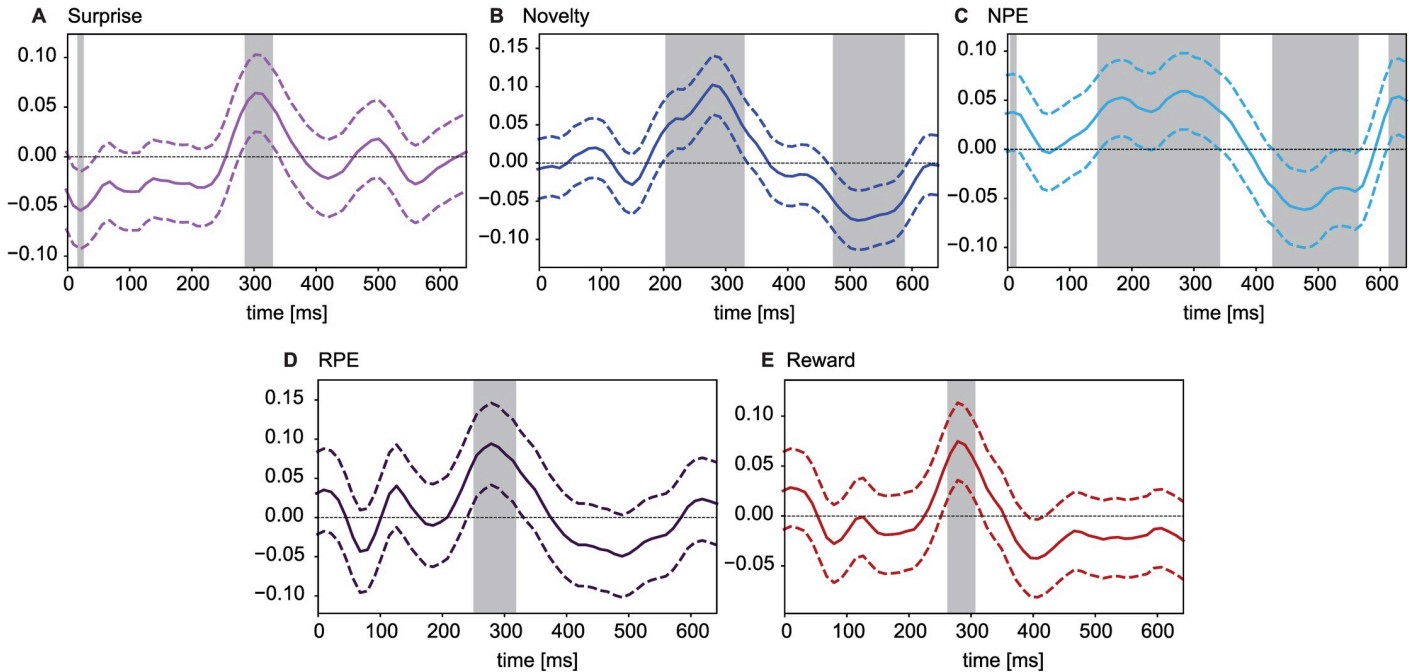

**Fig 9. Grand correlation analysis of normalized ERPs over all 2524 trials of 10 participant.** The dashed lines show confidence intervals. Shaded areas indicate intervals of significant correlations (FDR controlled by 0.1, one-sample t-test). Correlations of ERP with **A**. Surprise, **B**. Novelty, **C**. NPE, **D**. RPE (computed after excluding the trials from the 1st episode of the 1st block during which RPE is equal to 0) and **E**. Reward.

## EEG correlates with novelty and surprise

Since surprise and novelty turned out to be important and independent components of Sur-NoR in explaining the participants' behavior, we wondered whether they are both reflected in the ERP. We first performed a grand correlation analysis in which we pooled the more than 2500 trials of 10 participants together after normalizing their ERPs to unit energy (see Methods; two participants were excluded because of noise artifacts in the recordings). We then computed the correlations of the ERP amplitudes, for each time point after the trial onset, with the model variables 'Surprise', 'Novelty', 'Reward, 'NPE', or 'RPE' (capital initial letters indicate the 5 model variables). Note that by 'Reward' variable we mean the goal-state indicator, i.e., it is equal to one when a participant visits the goal state and zero otherwise; importantly, it should not be confused with MB or MF reward values.

We find that Surprise, Reward, and RPE show significant positive correlations with the ERP amplitudes at around 300ms after stimulus onset (Fig 9), in agreement with the well known correlation of the P300 amplitude with Surprise [23, 50, 56] and the well known correlation of the Feedback-Related Negativity (FRN) component with RPE [57, 58]. Moreover Novelty and NPE have, compared to Surprise, a broader positive correlation window with the ERP starting at around 200ms and ending at around 320ms after stimulus onset, and a second window with significant negative correlations from around 450ms to 550ms. Thus, Novelty and NPE have an ERP signature that is distinct from that of Surprise, Reward, or RPE (Fig 9).

Second, we wondered how much of the variations in the ERP amplitudes could be explained by a linear combination of our five model variables, i.e., Suprise, Novelty, NPE, RPE, and Reward. We performed a trial-by-trial multivariate linear regression (MLR), separately for each participant. To be able to more precisely identify the separate contributions of each model variable to the regression, we needed to decorrelate them from each other. As expected

from the design of the experiment, the cross-correlations between the normalized (zero mean and unit variance) sequences of Surprise, Novelty, and NPE are negligible (see S6 Fig); however, the sequences of Reward and RPE are highly correlated with each other, mainly because Reward and RPE are both high at the goal state. Using principal components analysis over Reward and RPE, we find $R_+$ (the sum of RPE and Reward) and $R_-$ (their difference) as their decorrelated combinations (see Methods and S2 Text). We then extracted the components of Surprise, Novelty, and NPE orthogonal to $R_+$ and $R_-$ (see Methods and S2 Text). The resulting variables, denoted by an index $\perp$, are each orthogonal to $R_+$ and $R_-$, while staying very similar to the original signals, e.g., Surprise$_\perp$ is highly correlated with Surprise, and NPE$_\perp$ is highly correlated with NPE; see S2 Text and S6 and S7 Figs for more details.

For each participant, we considered the normalized Surprise$_\perp$, Novelty$_\perp$, NPE$_\perp$, $R_+$, and $R_-$ as explanatory variables in order to predict the ERP amplitude at a given time point. We found 4 time intervals with an encoding power (adjusted R-squared, see Methods) significantly greater than zero (one-sample t-test, FDR controlled by 0.1, Fig 10A and 10B; note that the

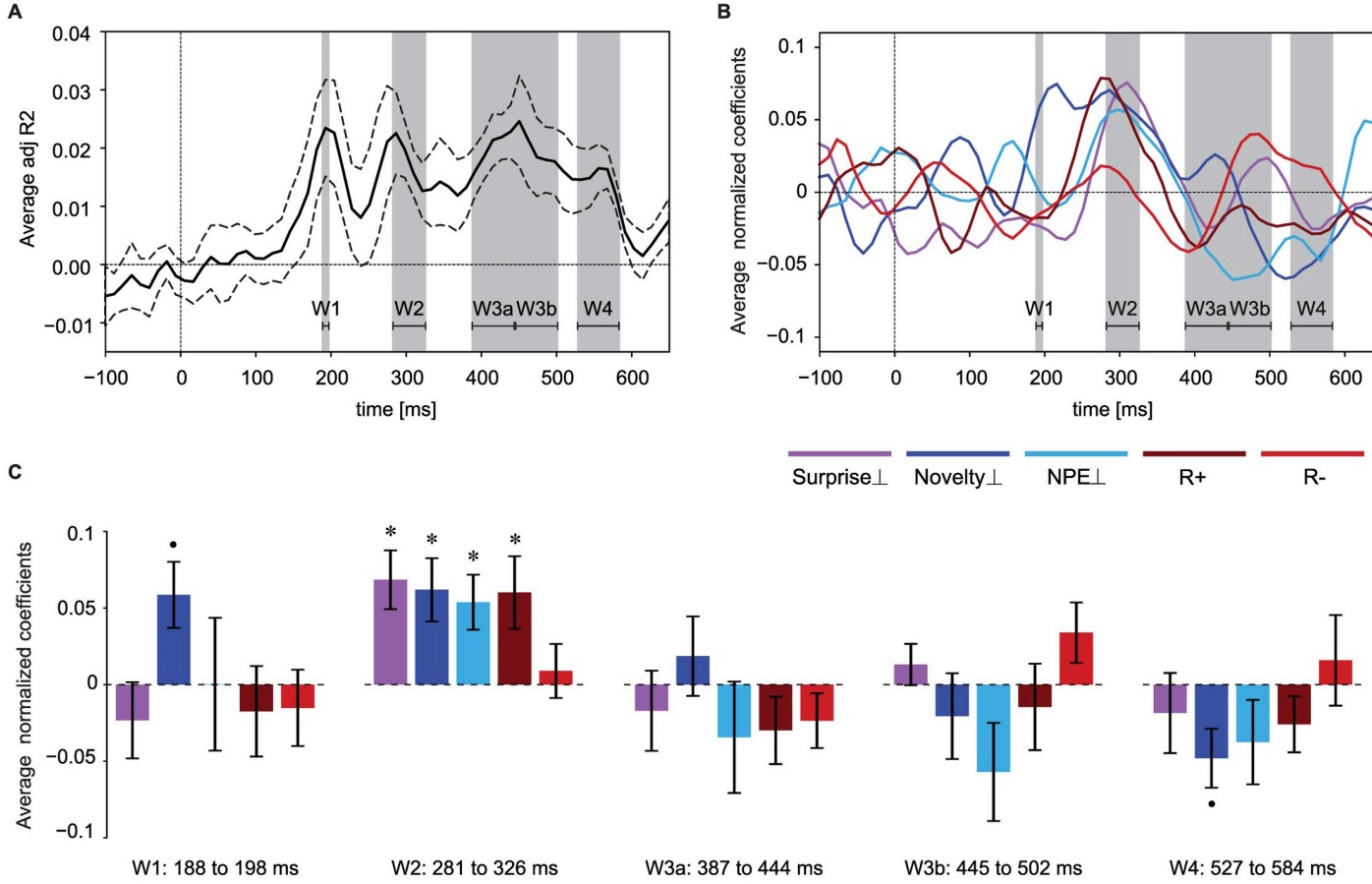

**Fig 10. ERP variations explained by trial-by-trial and participant-by-participant multivariate linear regression analysis.** Surprise$_\perp$ (magenta), Novelty$_\perp$ (dark blue), NEP$_\perp$ (light blue), $R_+$ (brown) and $R_-$ (red) were used as explanatory variables, and the ERP amplitude at each time point was considered as the response variable. **A**. Encoding power (adjusted R-squared values) averaged over 10 participants (dashed lines show the standard error of the mean) at each time point. Shaded areas and horizontal lines indicate four time intervals (W1, ..., W4) of significant encoding power (FDR controlled by 0.1, one-sample t-test, only for the time-points after the baseline). The 3rd time interval has been split into two time windows of equal length for the analysis in C. **B**. Values of the regression coefficients (averaged over participants) for Surprise$_\perp$, Novelty$_\perp$, NEP$_\perp$, $R_+$, and $R_-$ as a function of time. Errors are not shown to simplify the illustration. **C**. In each of the 5 time windows, the regression coefficients plotted in B have been averaged over time. Error bars show the standard error of the mean (across participants). Asterisks show significantly non-zero values (FDR controlled by 0.1 for each time window, one-sample t-test). The Novelty$_\perp$ coefficients in the 1st and the last time windows (dot) have p-values of 0.03 and 0.04, respectively, which are not significant after FDR correction. In the second time window, Surprise$_\perp$, Novelty$_\perp$, NEP$_\perp$, and $R_+$ have significantly positive coefficients.

adjusted R-squared can take negative values, e.g., see baseline in Fig 10A). The 1st time window is around 193 ± 5ms; the P300 component can be linked to the 2nd time window which spans from 286 to 321 ± 5ms; since the 3rd time interval is long (from 392 to 487 ± 5ms), we split it into two time windows of equal size (W3a and W3b in Fig 10B); and the last window extends from 532 to 574 ± 5ms.

To study the contribution of surprise and novelty to encoding power, we focused on these time windows and tested the average regression coefficients of all explanatory variables in each time window in a second level analysis (Fig 10C). Our results show that in the the second time window (286 to 321 ± 5ms), $\text{Surprise}_\perp$, $\text{Novelty}_\perp$, $\text{NPE}_\perp$, and $R_+$ all have a significant positive regression coefficient in MLR (Fig 10C, 2nd panel, FDR controlled by 0.1). While the coefficients for $\text{Surprise}_\perp$, $\text{NPE}_\perp$, and $R_+$ sharply peak at around 300ms, the coefficient for $\text{Novelty}_\perp$ has a broader peak starting at around 200ms (Fig 10B) with a close to significant positive value during the 1st time window (Fig 10C). This observation suggests that positive correlations of novelty with the ERP potentially extend from the 1st time window to the 2nd one, in agreement with our grand correlation analysis.

While consistent with previous studies of surprise in the ERP [22, 23, 50, 56], our results indicate that $\text{Surprise}_\perp$ and $\text{Novelty}_\perp$ contribute each separately to the ERP components at around 300ms. Furthermore, we find that $\text{NPE}_\perp$ is yet another independent contributor to these components. As expected from previous studies [57, 58], $R_+$ also shows a positive correlation with the ERP amplitude at around 300ms. While the multivariate analysis based on the 5 explanatory variables shows significance in the later time windows (Fig 10A), individual contributions of $\text{Surprise}_\perp$ or $\text{Novelty}_\perp$ or $\text{NPE}_\perp$ alone remain below significance level even though $\text{Novelty}_\perp$ has a close to significant negative coefficient in the last window (Fig 10C).

To summarize, the grand correlation analysis yields time windows of significance for Novelty and NPE that start 50 to 100ms *before* those of Surprise or Reward, indicating distinct contributions. Moreover, $\text{Novelty}_\perp$ and $\text{NPE}_\perp$ explain a significant fraction of the variations of the ERP at around 300ms that is not explained by $\text{Surprise}_\perp$ and $R_+$ alone. Importantly, NPE has significant correlations both in the grand correlation analysis and in the regression analysis, consistent with our earlier finding that NPE is important to explain behavior.

## Discussion

Combining a deep sequential decision-making task with the SurNoR model, an augmented reinforcement learning algorithm, we were able to extract the distinct contributions of surprise, novelty, and reward to human behavior. We found that the human brain (i) uses surprise to adapt their behavior to changing environments by modulating the learning rate and (ii) uses novelty as an intrinsic motivational drive to explore the world. Moreover, the model variables Suprise, Novelty, NPE, Reward, and RPE could well explain variations of the EEG amplitudes on a trial-by-trial basis.

### Novelty is not surprise

In the SurNoR model, surprise measures how *unexpected* the next state is according to an acquired world-model *conditioned* on the current state and the chosen action; in contrast, novelty measures how *infrequent* the next state has been, *independent* of our expectations derived from a world-model. More precisely, in our formulation (Eqs 2 and 4), surprise and novelty have two essential differences: The first difference is that while surprise is assigned to transitions, novelty is assigned to states. If this was the only difference between novelty and surprise, one could argue that surprise and novelty are essentially the same, while one measures the infrequency of transitions and the other the infrequency of states. However, the second and

more important difference is that novelty uses the exact number of encounters of each state (Eq 1) to measure how infrequent that state has been, while surprise uses the surprise-modulated pseudo-counts (Eq 3) computed by the world-model to measure how unexpected a transition is. As a consequence, if there is a sudden change in the environment, then an expected transition can rapidly become surprising, but a long time is needed for a state that has been encountered frequently to become novel again. This is consistent with ideas that novelty is more related to memory-recall and surprise is more related to predictions [35]. Moreover, these ideas are also supported by recent findings showing that the brain estimates the frequency of stimuli over a much longer time-scale than the transition probabilities [22].

Measures of surprise in neuroscience have been divided into two subgroups [28, 34, 39]: 'puzzlement' and 'enlightenment' surprise. The conceptual definition of surprise we gave above is known as puzzlement surprise. In addition to the Bayes factor surprise [29] that we used here (Eq 4), other examples of puzzlement surprise are Shannon surprise [46], minimized free energy [59, 60], state prediction error [5], and the confidence corrected surprise [28]. Orthogonal to these measures, enlightenment surprise measures how much an event changes our model of the world and, as a consequence, our expectations; well-known examples are Bayesian surprise [12, 61–64] and compression surprise [12, 63]. We would like to emphasize here that even a measure like Bayesian surprise [12, 61, 62, 64] has the aforementioned differences with our definition of novelty.

Measures of novelty can also be divided into two subgroups [35]: the ones that are 'memory-based' and the ones that represent 'statistical outliers'. Memory-based novelty measures focus on whether an event already exists in the memory or not [65, 66]. Our measure of novelty (Eq 2) belongs to the 2nd group that considers an event as novel if it has 'a low estimated probability of occurrence' [35]. Many measures of novelty that belong to this group simply consider novelty to be a decreasing function of the number of occurrences [9, 15, 17]; in contrast, our measure of novelty is a decreasing function of the *frequency* of occurrence (Eq 1), and because of this reason one may refer to it as 'relative novelty' or a measure of 'rareness'.

## Surprise modulates learning

As expected from previous theoretical [28–31, 33, 67] and experimental work [20, 24–26], our results suggest that the human brain uses surprise to modulate the learning of its world-model. Rather unexpectedly, however, our results indicate that humans hardly use this world-model to plan behavior; instead they mainly rely on model-free TD learning with eligibility traces to choose their next actions. Importantly, although the surprise signal is triggered by a mismatch between an observation and the predictions of the world-model, the modulatory effect of surprise is not limited to readjusting the world-model but also used to modulate the learning rate of model-free TD-learning. Following the common interpretations of model-based reinforcement learning algorithms as descriptions of human planning behavior and model-free reinforcement learning algorithms as descriptions of human habitual behavior [3, 5, 6, 68], our results suggest that (i) in the absence of surprise, humans prefer habitual behavior (potentially to reduce computational costs of decision-making [69, 70]) and (ii) errors in their world-model make them reconsider their habitual behavior.

Our results extend findings that humans use hybrid policies in two-stage decision tasks [5, 6] to the case of deep sequential decision tasks in the presence of abrupt changes. In general, the balance between model-free and model-based behaviors depends on multiple aspects and features of the task that humans are dealing with [3, 5, 70, 71]. For example, we observe that the model-based branch becomes more important when participants explore the environment to find the goal state, although the participants' behavior in our environment is always

dominated by model-free action choices (Fig 6A4). Moreover, the dominance of model-free behavior in such deep tasks does not exclude that humans used model-based planning in shallow tasks that are easily comprehensible thanks to a spatial arrangement of states or explicit instructions [72].

An interesting direction for future studies is to combine surprise modulation with more abstract model building algorithms, e.g., for learning the structure of neighborhood relations of an environment in the form of a graph [73, 74]. Such algorithms may explain the slight difference between the participants' adaptive behavior and SurNoR's predictions (Fig 6B).

## Novelty drives exploration

Our results show that exploration based on novelty-seeking can explain human behavior in our sequential decision-making task better than its alternatives: random exploration [75], optimistic initialization [19], and uncertainty or surprise-seeking [42, 43]. In contrast to many exploration strategies that give preference to those actions for which the outcome is most uncertain, i.e., those that have been tried least [42, 43, 75–77], exploration based on novelty-seeking gives preference to actions that ultimately lead a participant to previously unvisited or less visited states, even if the participant is perfectly sure about the transition to the next state.

In general, it has been shown that exploration in humans can have more than one drive [77] and that participants' desire for seeking novel events depends on their goal, inductive biases, and assumptions about their environment [9, 15, 37]. In situations like our experimental paradigm where participants are sure that there exists a rewarding state but do not know how to reach it, seeking novelty and exploring the parts of the environment that have been less visited are natural ways to search for the rewarding state; however, if, for example, participants were asked to find the most accurate map of the environment, then uncertainty-seeking might be a more reasonable way to solve the task. In addition, the presence of trap states in our environment makes novelty an internally rewarding signal that helps participants to avoid 'traps'. We do not claim that novelty is always the only drive of exploration; rather, we believe that our results show that for a class of tasks similar to ours where the goal is to search for a rewarding state and novelty is a relevant signal, humans use a novelty-seeking strategy for exploration. Following the idea of information search in active sampling [78, 79], we speculate that whether novelty is an informative cue (e.g., about the location of the goal) or not must be itself inferred by participants through the exploration procedure. From a different but similar perspective, we speculate that it is possible to take a normative approach and, by defining a function of curiosity [15, 37], show that novelty-seeking is the optimal or a close to optimal way to search for reward in a class of environments. Formulating and testing such hypotheses is an interesting direction for future studies.

The SurNoR algorithm suggests that participants treat novelty and reward as separately estimated values—as opposed to adding them into a single value estimator [10, 17, 18]. This separation enables participants to rapidly switch from exploration to exploitation, once they have found the goal. Based on this insight, we make the following prediction: if participants find a goal state but expect a second more rewarding goal state, they will continue to explore and potentially spend a large amount of time in a novelty-rich segment of an extended version of the environment of Fig 1 (see S5 Text).

## Neural signatures of surprise and novelty

Our EEG analysis shows that variables of the SurNoR model can significantly explain the variations of ERP amplitudes in several time-windows: Surprise, Novelty, NPE, and Reward/RPE all significantly contribute to the encoding power in the time-window around 300ms. The

positive contributions of Novelty, Surprise, and Reward/RPE in this time-window are consistent with previous studies of the P300 and the FRN component [22, 23, 38, 50, 56–58]. Whereas in earlier studies contributions of Novelty, Surprise, and Reward were often mixed together [22, 23, 38, 50, 56–58], we have shown here separate, additive contributions of these three variables as well as a further contribution of NPE. The effect of Novelty appears in ERPs earlier (at around 200ms) than the correlations with the other variables; moreover, contributions of Novelty are distinct from those of Surprise in the time window after 400ms.

Since, in the SurNoR model, Novelty is treated analogously to an external Reward, TD-learning based on NPE along with eligibility traces rapidly diffuses information about novel states to far-away non-novel states just as TD-learning based on RPE along with eligibility traces rapidly diffuses information about rewarding states to far-away non-rewarding states. Several studies have shown that the reward-driven activity of dopaminergic neurons encodes RPE and not reward values [1, 80, 81]. Therefore, the manifestation of a separate NPE signal in neural activities may open a new door for further developments of theories and experiments on novelty-driven activity of dopaminergic neurons and other neuromodulators [82–85].

## Conclusions

In conclusion, surprise and novelty are conceptually distinct concepts that also give rise to different temporal components in the ERP. Our results suggest that humans use novelty-seeking for efficient exploration and surprise for a rapid update of both their internal world-model and their model-free habitual responses.

## Methods

### Ethics statement

The data were collected under CE 164/2014, and the protocol was approved by the Commission cantonale d'éthique de la recherche sur l'être humain. All participants were informed that they could quit the experiment at any time, and they all signed a written informed consent. All procedures complied with the Declaration of Helsinki (except for pre-registration).

### Experimental setup

Stimuli were presented on an LCD screen that was controlled by a Windows 7 PC. Experiments were scripted in MATLAB using the Psychophysics Toolbox [86].

### Participants

14 paid participants joined the experiment. Two participants quit the experiment (14 ± 10% of all participants), hence, we analysed data for 12 participants (5 females, aged 20–26 years, mean = 22.8, std = 1.7). All participants were right-handed and naive to the purpose of the experiment. All participants had normal or corrected-to-normal visual acuity.

### Stimuli and general procedure

Before starting the experiment, we showed the participants the goal image that they were required to find on a computer screen. Next, participants were presented, in random order, all the other images that they might encounter during the experiment. Thereafter, participants clicked the 'start' button to start the experiment. At each trial, participants were presented an image (state) and four grey disks below the image. Clicking on one of the disks (action) led participants to a subsequent image; for details of timing see Fig 1A. Participants clicked through the environment until they found the goal state which finished the episode. An

episode $n$ started at a random state $i(n)$ which was the same for all participants; in our experiment we used $i(1) = 6$, $i(2) = 9$, $i(3) = 4$, $i(4) = 5$, and $i(5) = 8$.

## EEG recording and processing

EEG signals were recorded using an ActiveTwo Mk2 system (BioSemi B.V., The Netherlands) with 128 electrodes at a 2048Hz sampling rate. Two participants were excluded from EEG analysis because of their noisy and low quality signals caused by substantial movements during the experiment. Data were band pass filtered from 0.1Hz to 40Hz and down sampled to 256Hz. EEG data were recorded with a Common Mode Sensor (CMS) and re-referenced using the common average referencing method. We used EEGLAB [87] toolbox in MATLAB to perform the EEG preprocessing. We extracted EEG trials from 200ms before to 700ms after the state onset. Trials in which the change in voltage at any channel exceeded 35 $\mu$V per sampling point were discarded. Eye movements and electromyography (EMG) artefacts were removed by using independent component analysis (ICA). The baseline activity was removed by subtracting the mean calculated over the interval from 200ms to 0ms before the state onset. EEG data of selected prefrontal electrodes (Fz, F1, F2, AFz, FCz) were averaged for ERP analysis. We further smoothed (moving averaging with the window of length 50ms) and downsampled (to the sampling rate of 1 sample per 11.7ms) ERPs. Data were analyzed during the time window from 0 to 650ms after state onset (blue interval in Fig 1A). For multivariate regression analysis, a 100ms-baseline was also included for sanity check. As a result, each trial (from 100ms before to 650ms after the onset of the state) consisted of 65 time points.

## SurNoR algorithm

We present a more detailed formulation and the psudocode of the SurNoR algorithm in S1 Text. Here we outline the algorithm in brief.

We formally define the **Novelty** of a state $s$ at time $t$ as $N^{(t)}(s) = -\log p_N^{(t)}(s)$, where $p_N^{(t)}$ is defined in Eq 1; see S1 Text for further discussion. When observing the image corresponding to state $s_{t+1}$ at time $t + 1$, after taking action $a_t$, the novelty $n_{t+1} = N^{(t)}(s_{t+1})$ is treated as an internal novelty-reward, completely analogous to the treatment of external rewards in reinforcement learning. This analogy between external reward and novelty is inspired by earlier experimental studies [82, 83, 88]. As a result, at time $t + 1$, agents receive three signals: the next state $s_{t+1}$, the external reward $r_{t+1}$ (i.e., the indicator of whether $s_{t+1}$ is the goal state or not), and the novelty $n_{t+1}$ (indicated as the output of the grey block in Fig 4C).

The SurNoR algorithm has two branches, i.e., a model-based and a model-free one, which interact with each other (Fig 4C, blue and red blocks). The **model-based branch** computes the Bayes Factor **Surprise** [29]

$$\mathbf{S}_{\mathrm{BF}}^{(t+1)} = \frac{p_{\mathrm{reset}}(s_{t+1}|s_t, a_t)}{p^{(t)}(s_{t+1}|s_t, a_t)} \tag{5}$$

where $p^{(t)}(s_{t+1}|s_t, a_t)$ is the probability of observing $s_{t+1}$ by taking action $a_t$ in state $s_t$ as estimated from the current world-model (cf., Eq 3), and $p_{\mathrm{reset}}(s_{t+1}|s_t, a_t)$ is the probability of observing $s_{t+1}$ by taking action $a_t$ in state $s_t$ with the assumption that the environment has experienced an abrupt change between time $t$ and $t + 1$, so that the world-model should be reset to its prior estimate. In this work, we assume that the prior estimate $p_{\mathrm{reset}}(s_{t+1}|s_t, a_t) = 1/11$ is a uniform distribution over states and hence constant as stated in Eq 4; see S1 Text and [29] for further discussion. Note that in Figs 4A, 4B and 6 we suppressed the factor 1/11 and directly plotted $1/p^{(t)}(s_{t+1}|s_t, a_t)$ as the surprise value. As an aside we note that since the state prediction error [5] is defined as $SPE_{t+1} = 1 - p^{(t)}(s_{t+1}|s_t, a_t)$, the Bayes Factor Surprise can be

written as $\mathbf{S}_{\text{BF}}^{(t+1)} \propto 1/(1 - SPE_{t+1})$. The definition of the Bayes Factor Surprise is valid for arbitrary volatile environments [29]. However, since in our experimental setting $p_{\text{reset}}(s_{t+1}|s_t, a_t)$ is assumed to be uniform, the Bayes Factor Surprise $\mathbf{S}_{\text{BF}}$ is a monotone function of Shannon Surprise and hence comparable to previous studies [23, 46, 50].

The value $\mathbf{S}_{\text{BF}}^{(t+1)}$ is used in the model-based branch to update the world-model using the Variational SMiLe algorithm [29], an approximate Bayesian learning rule with surprise-modulated learning rate designed for volatile environments with abrupt changes. Updating the world-model is equivalent to updating the pseudo-counts $\tilde{C}_{s,a\rightarrow s'}^{(t)}$, introduced in Eq 3, for all possible $s$, $a$, and $s'$. The Variational SMiLe algorithm [29] yields the updates

$$\tilde{C}_{s,a\rightarrow s'}^{(t+1)} = \begin{cases} (1 - \gamma_{t+1})\tilde{C}_{s,a\rightarrow s'}^{(t)} + \delta(s', s_{t+1}) & \text{if} \quad s = s_t, a = a_t \\[2mm] \tilde{C}_{s,a\rightarrow s'}^{(t)} & \text{otherwise,} \end{cases} \tag{6}$$

where $\delta$ is the Kronecker delta function, and $\gamma_{t+1}$ is the surprise modulated adaptation factor [29]

$$\gamma_{t+1} = \frac{m\mathbf{S}_{\text{BF}}^{(t+1)}}{1 + m\mathbf{S}_{\text{BF}}^{(t+1)}} \in [0, 1], \tag{7}$$

with $m \geq 0$ a free parameter related to the volatility of the environment [29]. Note that if the transition from $s$ to $s'$ caused by action $a$ is unsurprising, then the pseudo-count of that transition is increased by one (because $\gamma = 0$ for $\mathbf{S}_{\text{BF}} = 0$). However, if this transition has a high surprise, the earlier pseudo-count is reset to zero (because $\gamma \rightarrow 1$ for $\mathbf{S}_{\text{BF}} \rightarrow \infty$) and the observed transition is counted as the first one. The updated world-model is then used to update a pair of $Q$-values, i.e., $Q_{\text{MB,R}}^{(t+1)}$ for Reward and $Q_{\text{MB,N}}^{(t+1)}$ for Novelty, by solving the corresponding Bellman equations with a variant of prioritized sweeping [19, 89, 90]; see S1 Text for details.

The **model-free branch** computes Reward and Novelty prediction errors, $RPE_{t+1}$ and $NPE_{t+1}$. As usual, RPE is defined as $RPE_{t+1} = r_{t+1} + \lambda_R V_{\text{MF,R}}(s_{t+1}) - Q_{\text{MF,R}}(s_t, a_t)$, where $\lambda_R$ is the discount factor for reward, and $V_{\text{MF,R}}(s_{t+1}) = \max_a Q_{\text{MF,R}}(s_{t+1}, a_t)$ is the value of the state $s_{t+1}$. Analogously, NPE is defined as $NPE_{t+1} = n_{t+1} + \lambda_N V_{\text{MF,N}}(s_{t+1}) - Q_{\text{MF,N}}(s_t, a_t)$, where $\lambda_N$ is the discount factor for novelty, and $Q_{\text{MF,R}}$ by $Q_{\text{MF,N}}$ is the novelty value of the state $s_{t+1}$.

A Surprise-modulated TD-learner with eligibility traces is used for updating the two separate sets of $Q$-values. To have the most general setting, two separate eligibility traces are used for the update of $Q$-values, one for reward $e_R^{(t)}$ and one for novelty $e_N^{(t)}$. The eligibility traces are initialized at zero at the beginning of each episode. The update rules for the eligibility traces after taking action $a_t$ at state $s_t$ is

$$\begin{aligned} e_R^{(t+1)}(s, a) &= \begin{cases} 1 & \text{if} \quad s = s_t, a = a_t \\[2mm] \lambda_R \mu_R e_R^{(t)}(s, a) & \text{otherwise} \end{cases} \\[4mm] e_N^{(t+1)}(s, a) &= \begin{cases} 1 & \text{if} \quad s = s_t, a = a_t \\[2mm] \lambda_N \mu_N e_N^{(t)}(s, a) & \text{otherwise}, \end{cases} \end{aligned} \tag{8}$$

where $\lambda_R$ and $\lambda_N$ are the discount factors defined above, and $\mu_N \in [0, 1]$ and $\mu_R \in [0, 1]$ are the decay factors of the eligibility traces for novelty and reward, respectively. The update rule is then $\Delta Q_{\text{MF}}^{(t+1)}(s, a) = \rho_{t+1} e^{(t+1)}(s, a) PE_{t+1}$, where $e^{(t+1)}$ is the eligibility trace (i.e., either $e_R^{(t+1)}$ or

$e_N^{(t+1)}$), $PE_{t+1}$ is the prediction error (i.e., either $RPE_{t+1}$ or $NPE_{t+1}$) and

$$\rho_{t+1} = \rho_b + \gamma_{t+1}\delta\rho \qquad (9)$$

is the surprise-modulated learning rate with parameters $\rho_b$ for the baseline learning rate and $\delta\rho$ for the effect of Surprise.

Finally, actions are chosen by a hybrid policy (S1 Text) using a softmax function of a linear combination of the values $Q_{MF,R}^{(t+1)}$, $Q_{MF,N}^{(t+1)}$, $Q_{MB,R}^{(t+1)}$, and $Q_{MB,N}^{(t+1)}$ (the purple block in Fig 4A), similar, but not identical to [5, 6]. The weight of $Q_{MF,N}^{(t+1)}$ and $Q_{MB,N}^{(t+1)}$ is non-zero only in the 1st episodes of blocks 1 and 2 to reduce number of parameters and make the model simpler. We tested the version with an additional free parameter for the weights of $Q_{MF,N}^{(t+1)}$ and $Q_{MB,N}^{(t+1)}$ in episodes 2–5 of blocks 1 and 2, but we did not find any significant improvement in the fit (difference in log-evidence = 15 ± 13).

Overall, the SurNoR algorithm has 18 free parameters.

## Statistical model analysis and fit to behavior

In addition to SurNoR, we considered 12 alternative algorithms with 0 to 18 free parameters and two control algorithms for SurNoR with Binary Novelty with 19 free parameters (S1 Text). For each algorithm, we used 3-fold cross-validation and computed its maximum log-likelihood for each participant, similar to existing methods [8]: (i) we divided participants into 3 folds each consisting of four participants; (ii) for participant *i*, we estimated the parameters of the algorithm by maximizing the likelihood function of the folds which did not include participant *i*; and (iii) we computed the log-likelihood for participant *i* using the estimated parameters. The maximization procedure was done by coordinate ascent (using grid search for each coordinate); we repeated the procedure until convergence starting from 25 different random initial points. We further repeated the whole process 4 times to have an estimation of the variability resulting from random initialization of the optimization procedure. The error bars in Fig 5A are calculated using these 4 samples.

Similar to studies in economics and statistics [91, 92], we considered, for each participant and each algorithm, the cross-validated maximum log-likelihood (averaged over the 4 repetitions) as the log-evidence [93]. Similarly, the log-evidence could also be approximated by other measures like AIC or BIC [94], but cross-validation has been shown to have a more robust behavior [95]. The sum (over participants) of the log-evidences for each algorithm is shown in Fig 5A—see [94] for a tutorial on the topic. As a convention, differences greater than 3 or 10 are considered as significant or strongly significant, respectively [93, 94]. The model posterior and protected exceedance probabilities in Fig 5B are computed by using the participant-wise log-evidences (averaged over the 4 repetitions) and following the Bayesian model selection method of [52, 53] (available in SPM12 toolbox for MATLAB). We used a Dirichlet distribution with parameters equal to 1 over the number of models (1/13) as the prior distribution. This choice of prior is equivalent to stating that the prior information is worth as much as the observation coming from a single participant [93].

The accuracy rate and the uncertainty in Fig 5C are computed by the same cross-validation procedure. We define accuracy as the ratio of the number of trials with correctly predicted actions to the total number of trials; for a given trial, whenever the action taken by the participant had the maximum probability under the policy but shared with other $n-1$ (e.g., 2) actions, we counted that trial as $1/n$ (e.g., 0.333) correctly predicted. With this procedure, the accuracy rate of the random choice algorithm is 25%. We define the uncertainty of one

participant in an episode as the average of the entropy of his or her policy over all trials of that episode. Both the accuracy rate and the uncertainty were computed for each participant separately, but only the mean and the standard error across participants are reported in Fig 5C.

For EEG analysis, we only considered the SurNoR algorithm (i.e., the winner of statistical model selection). To have the same set of parameters for all participants, we fitted our model to the whole behavioral data set (overall 3047 actions) by maximizing total log-likelihood—similar to [6]. For each of 500 random initialization points, maximization was implemented as coordinate ascent until convergence (using grid search for each coordinate). Amongst the 5 local maxima with high but not significantly ($< 3$) different log-evidence, we kept the model which had the greatest encoding power in multivariate regression analysis of EEG. The fitted parameters are reported in S3 Text and S1 Table.

The plots in Fig 6 corresponds to this set of parameters. Since the softmax operator of the hybrid policy has a free scale parameter, the effective weight of each branch of the hybrid policy in Fig 6A4 (i.e., model-free and model-based) is computed as the fitted weight of each component times its average difference in Q-values. For example, $\omega_{MF}^{\text{eff}}$ is equal to $\omega_{MF} \times \langle \Delta Q_{MF} \rangle$, where $\omega_{MF}$ is the weight of model-free Q-values in the hybrid policy and $\langle \Delta Q_{MF} \rangle$ is the average (over trials) of the difference between $Q_{MF}$ of the best and the worst actions. The weight $\omega_{MB}^{\text{eff}}$ for the model-based branch is defined analogously. The dominance of the model-free branch is defined as $\omega_{MF}^{\text{eff}} / \omega_{MB}^{\text{eff}}$.

We used the same set of parameters to generate synthetic data for Fig 7. We simulated 200 agents with different random seeds. We considered the 62 agents ($31 \pm 3\%$ of all agents) who took more than 500 actions in any of the 10 episodes to be similar to the participants who quit the experiments ($14 \pm 10\%$ of all participants—not significantly different from $31 \pm 3\%$; p-value for two-sample t-test = 0.12). Based on this criterion, we discarded 62 agents. From the remaining 138 agents, we randomly chose three subsets of 12 agents (called simulated participants in the Results section) and repeated all our behavioral analyses for the synthetic data. The results for one subset of agents is shown in Fig 7 and for two other subsets in S2 and S3 Figs. Given the three sets of 12 simulated agents, the results for model recovery is shown in Fig 8, and the results for parameter recovery are reported in S3 Text and S4 and S5 Figs.

## EEG analysis

**Participant-based regression analysis.**   Given $N$ trials (across all episodes of both blocks) of a given participant, the matrix $X_{\text{raw}}$ for this participant is an $N$ by 5 matrix whose rows correspond to trials and whose columns correspond to normalized model variables (i.e., Surprise, Novelty, NPE, RPE, and Reward). For example, if the sequence of reward prediction error values for this participant is $z_{1:N}$, then one column of the matrix $X_{\text{raw}}$ is equal to $(z_{1:N} - \mu_z)/\sigma_z$ where $\mu_z$ is the mean and $\sigma_z$ is the standard deviation of $z_{1:N}$, and one row of the matrix $X_{\text{raw}}$ is equal to the normalized values of Surprise, Novelty, NPE, RPE, and Reward for one trial. We constructed the feature matrix $X$ from $X_{\text{raw}}$ by applying the following steps: (i) we put 2 columns of $X$ to be equal to normalized Reward plus RPE and Reward minus RPE, calling them $R_+$ and $R_-$, respectively; since Reward and RPE were normalized, their sum and difference correspond to their principal components (S2 Text); (ii) we orthogonalized each of the other variables to $R_+$ and $R_-$. For example, $NPE_\perp$ is NPE minus its projection on $R_+$ and $R_-$, followed by a renormalization step (see S2 Text and S6 Fig).

For each trial, time of the ERP is measured with respect to the image onset. For a given time point, we defined the target vector $y$ as an $N$ dimensional vector whose elements are equal to the normalized (zero mean, unit variance) amplitude of ERPs at that particular time point in different trials. Since we have 65 time points, the response matrix $Y$ is a $N$ by 65 matrix. We

then performed multivariate linear regression (MLR) by considering $\hat{y} = X\beta$ as an estimation of $y$ and found $\beta$ by ordinary least squared error minimization. The encoding power for a single time point and for the given participant was calculated as adjusted R-squared [97]. Note that adjusted R-squared can in principle be negative—which is the case for our regression analysis over baseline in Fig 10A.

Fig 10A shows the mean and the standard error of the mean of the encoding power over participants and for each time-point. The threshold for rejecting the null hypothesis is computed using the Benjamini and Hochberg algorithm [93] for controlling false discovery rate (FDR) by 0.1. Fig 10B shows the average (over participants) of $\beta$ values as a function of time. For Fig 10C, we first average the $\beta$ values over time within each time window, and then evaluate their mean and their standard error of the mean (over participant). The FDR correction was done separately for each time window.

**Grand correlation analysis.**   Similar to the approach of [98], we pooled all trials of all participants together, i.e., we concatenated $X_{\text{raw}}$s and $Y$s for different participants. However, before concatenation, to remove the difference in the between-participant variations of ERPs energy (i.e., 2nd moment), we divided ERPs of each participant by the overall squared-energy of that participant's ERPs, i.e., we replaced $Y$ by $Y/\sqrt{\mathbb{E}[Y^2]}$. The correlations in Fig 9 are computed between columns of concatenated $X_{\text{raw}}$s and concatenated $Y$s. For RPE, we removed the trials corresponding to 1st episodes of the 1st blocks because RPEs are equal to zero.

## Supporting information

**S1 Text. SurNoR and alternative algorithms.**
(PDF)

**S2 Text. EEG preprocessing and control analyses.**
(PDF)

**S3 Text. Fitted parameters and parameter-recovery.**
(PDF)

**S4 Text. The analysis of random exploration.**
(PDF)

**S5 Text. Precise statement of the prediction in 'Discussion'.**
(PDF)

**S6 Text. Qualitative differences between model-based (MB) and model-free (MF) branches of SurNoR.**
(PDF)

**S1 Fig. Average progress at states in proximity of the goal (complement to Figs 2 and 7 of the main text).** Average progress of participants (**A**) and simulated participants (**B**-**D**) each time visiting states 5, 6, and 7 in episode 1 of block 1 (blue) or states 5, 6, and 3 in episode 1 of block 2 (red). Panel **A** corresponds to the experimental data shown in Fig 2, panel **B** to the simulated data shown in Fig 7, panel **C** to the simulated data shown in S2 Fig, and panel **D** to the simulated data shown in S3 Fig. While states 1 and 2 are visited by most participants at least 15 times (c.f. Fig 2, main text; size of circles indicates number of participants), only very few participants visit the states close to the goal more than 5 times, and as a result, total learning progress between the start and the end of the episode is smaller and data is noisy. Similar observations can be made for the simulated participants (cf. Fig 7, main text; size of circles indicates number of simulated participants). Since the number of visits of states close to the

goal is small, the noise-induced differences between different simulations runs are large (compare the runs in **B**, **C**, and **D**). In particular, at the state before the goal state (state 7 in block 1 and state 3 in block 2), the first time that (simulated) participants choose the good action, they reach the goal state and finish the episode. Therefore, the average progress for the state before the goal is always calculated across those (simulated) participants who did not choose the good action when they visited that state previously.
(EPS)

**S2 Fig. Posterior predictive checks; the structure of figure is the same as Fig 7 and their only difference is in the random seed used for generating data. A**. Average number of actions of 12 simulated participants at each episode (c.f. Fig 1C). **B**. Median number of actions of simulated participants to escape the trap states at each of their visits in episode 1 of block 1 (left) and block 2 (right) (c.f. Fig 2A) **C**. Average progress of participants each time visiting states 1, 2, 3, and 4 in episode 1 of block 1. (c.f. Fig 2B). **D**. Average progress of simulated participants each time visiting states 1, 2, 7 (swapped with 3), and 4 in episode 1 of block 2. (c.f. Fig 2C). See S1C Fig for the average progress at the progressing states in the proximity of the goal.
(EPS)

**S3 Fig. Posterior predictive checks; the structure of figure is the same as Fig 7, and their only difference is in the random seed used for generating data. A**. Average number of actions of 12 simulated participants at each episode (c.f. Fig 1C). **B**. Median number of actions of simulated participants to escape the trap states at each of their visits in episode 1 of block 1 (left) and block 2 (right) (c.f. Fig 2A) **C**. Average progress of participants each time visiting states 1, 2, 3, and 4 in episode 1 of block 1. (c.f. Fig 2B). **D**. Average progress of simulated participants each time visiting states 1, 2, 7 (swapped with 3), and 4 in episode 1 of block 2. (c.f. Fig 2C). See S1D Fig for the average progress at the progressing states in the proximity of the goal.
(EPS)

**S4 Fig. Parameter recovery results and log-likelihood landscape.** The solid black curve in each panel shows the log-likelihood of the behavioral data of 12 participants as a function of one of the free parameters of SurNoR, while the other parameters are fixed at their fitted values (S1 Table). The fitted value for each parameter corresponds to the peak of the log-likelihood function and is specified by the light green lines. The recovered parameters for 3 different sets of 12 simulated participants (corresponding to the data shown in Fig 7 in the main text and S2 and S3 Figs) are shown by the light red lines. Note that the procedure of fitting parameters to the generated data were exactly the same as the procedure of fitting parameters to the real data, i.e., we searched in the 18-dimensional space of parameters. The 1st column corresponds to parameters mainly related to model-building and model-based planning ($\epsilon$, $m$, and $T_{\mathrm{PS}}$); the 2nd column corresponds to the softmax policy temperatures and the parameters controlling the exploration and exploitation trade-off ($\beta_1$, $\beta_2$, $\beta_{N1}$, and $\beta_{N2}$); the 3rd column corresponds to the discount factors and the decay rates of eligibility traces ($\lambda_R$, $\lambda_N$, $\mu_R$, and $\mu_N$); the 4th column corresponds to parameters mainly related to model-free learning ($\rho_b$, $\delta\rho$, and $Q_{N0}$); and the last column corresponds to the parameters controlling the trade-off between model-based and model-free policies ($\omega_0$, $\omega_{11}$, $\omega_{12}$, and $\omega_{\mathrm{scale}}$). See S3 Text for details.
(EPS)

**S5 Fig. Robustness of model-variables.** Average correlation between the model-variables extracted given the fitted parameters (the ones used for EEG analyses) and model-variables extracted given the recovered parameters corresponding to **A**. Fig 7 in the main text, **B**. S2 Fig, and **C**. S3 Fig. Error bars show the standard error of the mean, and each grey point shows data

of one participant. See S3 Text for details.
(EPS)

**S6 Fig. Correlations (averaged over participants) between relevant variables. A**. The cross-correlations between Surprise, Novelty, NPE, RPE, and Reward during the behavioral task. **B**. Novelty$_\perp$ is the projection of Novelty onto the subspace orthogonal to the plane spanned by Reward and RPE. The variables $R_+$ and $R_-$ are the (normalized) sum and difference of RPE and Reward, respectively. An analogous orthogonalization is applied to Surprise and NPE. **C**. The cross-correlation matrix of the orthogonalized variables and the original ones. Surprise$_\perp$, Novelty$_\perp$, and NPE$_\perp$ are highly correlated with their raw values but have zero correlation with reward and RPE. See S2 Text for details.
(EPS)

**S7 Fig. ERP variations explained by trial-by-trial and participant-by-participant multivariate linear regression analysis.** S7 Fig uses a simplified preprocessing pipeline without orthogonalization but is otherwise analogous to Fig 10 in the main text. Surprise (magenta), Novelty (dark blue), NEP (light blue), Reward (brown) and RPE (red) were used as explanatory variables, and the ERP amplitude at each time point was considered as the response variable. **A**. Encoding power (adjusted R-squared values) averaged over 10 participants (dashed lines show the standard error of the mean) at each time point. Shaded areas and horizontal lines indicate four time intervals (W1, . . ., W4) of significant encoding power (FDR controlled by 0.1, one-sample t-test, only for the time-points after the baseline). The 3rd time interval has been split into two time windows of equal length for the analysis in C. **B**. Values of the regression coefficients (averaged over participants) for Surprise, Novelty, NEP, Reward, and RPE as a function of time. Errors are not shown to simplify the illustration. **C**. In each of the 5 time windows, the regression coefficients plotted in B have been averaged over time. Error bars show the standard error of the mean (across participants). Asterisks show significantly non-zero values (FDR controlled by 0.1 for each time window, one-sample t-test). The Novelty coefficients in the 1st and the last time windows (dot) have p-values of 0.03 and 0.04, respectively, which are not significant after FDR correction. In the second time window, Surprise, Novelty, and NEP have significantly positive coefficients. See S2 Text for details.
(EPS)

**S1 Table. SurNoR parameters fitted to the behavioral data of all participants.** This set of parameters was used for EEG analysis and illustrations in Fig 6 in the main text. The values correspond to the light green lines in S4 Fig. See S3 Text for details.
(XLSX)

## Acknowledgments

AM thanks Johanni Brea and Vasiliki Liakoni for useful discussions on behavioral modeling and data analysis.

## Author Contributions

**Conceptualization:** He A. Xu, Alireza Modirshanechi, Marco P. Lehmann, Wulfram Gerstner, Michael H. Herzog.

**Data curation:** He A. Xu.

**Formal analysis:** He A. Xu, Alireza Modirshanechi, Marco P. Lehmann.

**Funding acquisition:** Wulfram Gerstner, Michael H. Herzog.

**Investigation:** He A. Xu, Alireza Modirshanechi.

**Methodology:** Alireza Modirshanechi.

**Software:** He A. Xu, Alireza Modirshanechi, Marco P. Lehmann.

**Supervision:** Wulfram Gerstner, Michael H. Herzog.

**Validation:** Alireza Modirshanechi.

**Visualization:** Alireza Modirshanechi.

**Writing – original draft:** He A. Xu, Alireza Modirshanechi, Wulfram Gerstner, Michael H. Herzog.

**Writing – review & editing:** Alireza Modirshanechi, Wulfram Gerstner, Michael H. Herzog.

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
