## [Decision Letter · Decision Letter 0]

21 Feb 2021

Dear Mr. Modirshanechi,

Thank you very much for submitting your manuscript "Novelty is not Surprise: Human exploratory and adaptive behavior in sequential decision-making" for consideration at PLOS Computational Biology.

As with all papers reviewed by the journal, your manuscript was reviewed by members of the editorial board and by several independent reviewers. In light of the reviews (below this email), we would like to invite the resubmission of a significantly-revised version that takes into account the reviewers' comments.

I would like to amplify a few points made by the reviewers:

- The evidence for the winning model comes mainly from model comparisons. It is important to show through model-agnostic analysis and predictive checks what exactly the winning model is capturing which the other models are not.

- 12 subjects is far too few for this kind of study, particularly involving EEG, which has a relatively low signal-to-noise ratio. Given the current circumstance, I won't ask you to collect more data (though that would of course be welcome), but you need to demonstrate using model recovery that your experimental setup is in principle capable of supporting your conclusions.

- Some of the models are very complicated and have many free parameters. This is not a problem in general, but my concern is that you are drawing rather strong conclusions about your model space from a specific task that is very simple compared to the complexity of the models. There's no way for your task to tease apart the roles of all the different parts of the models. It would build confidence that you're not overfitting if you were able to show that the winning model could also capture data from other sequential decision tasks. There are multiple publicly available data sets that you could analyze. However, I will not make this a requirement of the revision if you can demonstrate at least through simulation that the model produces sensible behavior on other tasks.

- The definition of "novelty" is problematic, since (as pointed out by one of the reviewers) it implies that a frequently experienced stimulus could be defined as novel as long as other stimuli are even more frequently experienced. This defies common sense. Essentially this is a terminological issue.

We cannot make any decision about publication until we have seen the revised manuscript and your response to the reviewers' comments. Your revised manuscript is also likely to be sent to reviewers for further evaluation.

Sincerely,

Samuel J. Gershman

Deputy Editor

PLOS Computational Biology

Reviewer's Responses to Questions

**Comments to the Authors:**

Reviewer #1: In the current study, individuals completed a sequential decision task with two unique features. First, the task included 'trap' trials, creating more visits in some states than others. Second, at the second half of the session, two states were swapped, creating transitions that deviated from the transition probability the individuals learned in the first part of the experiment. These two elements in the task design are used to test whether individuals are using 'state-novelty' (i.e., related to number of visits in a state), and 'transition-supervise' (related to getting to an unexpected state) to direct their actions. The authors describe in detail a hybrid model where novelty-seeking serves as a motivational goal, and transition-surprise modulates learning-rates. The authors claim that a hybrid model with model-based model-free, novelty and reward learning overcomes as models. The authors further argue using EEG data for distinct correlation between these two elements (i.e., 'state-novelty' and 'transition-surprise') with neural activity.

I would like to thank the authors for their hard work, which I believe is very interesting and provide further insights into action selection in humans. I think the manuscript enjoys both a details computational section, empirical model comparison and a section with neuronal activity correlations. In that sense, I believe the manuscript is well within the aims and scope of the current journal. I for have one major comment regarding model-selection that I will detail below. I hope this will serve the authors in improving their work, and I would like wish good luck and further fruitful studies.

Yours,

Nitzan Shahar

Sagol school of Neuroscience

Tel-Aviv University

Major Comment:

My major concern regards the fact that the model comparison includes a large amount of models (13 models), the winning model has a large amount of parameters (18 parameters), yet the data set is small (12 participants with 250 trials each). I would not a priori say it is impossible to make conclusions based on such setting, but I think we need to make sure the model selection method is justified, and that both the winning model and the parameter estimates are recoverable in this setting. The authors pooled data across three groups of subjects (4 subjects each), and used cross-validated maximum log-likelihood and protected exceedance probabilities for model selection. I think the model selection evidence will be much more compelling if the authors could show that: (a) the parameters of the winning model (which other than one model have the largest number of parameters in the model space) can be recovered in a simulation with the same number of subjects and trials. (b) show that the true model can be accurately recovered in this setting and with the same number of subjects, trials and pooling method. That is, simulate data and show we can accurately identify which model generated the data from all the relevant models. If the authors have in mind any other way to address parameter and model recoverability, that would be great as well.

------------------

Minor comments:

Line 495: The weight of Q(t+1) MF,N and Q(t+1)MB,N is non-zero only in the 1st episodes of blocks 1 and 2.

If I read this correctly, it means the author guides the model as to where 'novelty matters', affectively muting novelty prediction errors in blocks where novelty was assumed lower by design. I think it would be better to explain what happens when this is not done and whether the model comparison results are changed when novelty prediction errors are not muted in the whole task. I would expect the model to be able to win even with stricter conditions where novelty prediction errors take part across the whole task.

Line 255: We find that the model-free branch dominates the actions. Thus the world-model is of secondary importance for action selection and is mainly used to detect surprising events.

Given this conclusion, I would assume that the advantage of a hybrid model over a model-free only model comes mostly from few trials in the first episode of the second block, where surprise is manipulated experimentally. Yet the SurNoR model seems to win even when only behavior from the first block is used (Figure 4A). Why is that? If really the whole purpose of the model-based branch is to direct learning rate in these cases, wouldn't a simpler algorithm manipulating model-free learning rates (e.g., Pearce-Hall associability) lead to similar results with less parameters? I think the authors should at least mention these options, even if they are not fitted to the data directly.

Lines 72 – 79

These lines are important- but very hard to follow. Please try to explain what you mean by episode and block, and how is it possible that individuals performed different episodes without "knowing" they started from the beginning (line 72)? It took this reader a few repetitions until I understood the design.

Reviewer #2: This is a very intriguing paper on how humans explore and learn a complex state space in search for reward. The authors developed a complicated sequential decision-making task, in which the states are connected in a complicated and recurring way, also giving rise to “trap” states, which move them back to the beginning. Without a generative model, this state space is hard to learn and lots of transitions need to be explored in order to find n efficient way to the reward. This is shown by the very large number of steps taken in the first episode of every block. The authors then proceed to develop and novel reinforcement learning model that includes and estimate of the novelty and surprise (defined in a precise mathematical way) and they undertake a comprehensive model selection procedure to conclude that there new model outperforms all the different alternative (and mostly degenerate versions) of their model. They also find an unexpected result, namely that humans do not seem to rely on their world model that they build as they are learning about the state space - as demonstrated by the relative importance of MF over MB learning.

The paper is well-written and suited for the PLoS CB readership. I think presentation of the different model variants in the exhaustive model comparison could be somewhat improved, if the modular nature of (a) MB/MF/Hyb, (b) S/noS, (c) OI/U/N) and their systematic combination into different model variants could be made a bit more explicit.

Furthermore, I have two related suggestions for improvement of the paper.

1. The (model-free) analysis of the highly interesting exploration phase of the first episode in each block strikes me a really sparse. I only noticed the number of step take in each episode as a model-free measure of these data. However, it would be much more informative (and actually crucial to my second suggestion) to develop a few indices that would characterize the explorative behavior in this stage. For instance, on could think of the (mean) step number, after which the subjects have learned to avoid one of all of the trap states. Or the step number, after which they have learned to avoid the “recurring” action, which brings then back to the current state. Or one could define a kind of “exploration-exploitation index” fro each subject and each state, which ist the number of exploratory actions taken in each state before they settle on an exploitative (and henceforth constant) policy in each state. I am sure there are more indices that one could think of to describe the explorative behavior in this crucial first episode.

2. Having defined such indices above, one could use these as targets for an appropriate posterior predictive check of the SurNoR model. So far, I don’t think the paper includes a proper PPC - in my reading of the paper, the accuracies shown in Fig 4c refer to the experimental data that was also used to fit the model, but the fitted model was not used to generate new synthetic data that can be analyzed w.r.t to the indices defined above. Such an analysis would also underline that the SurNoR model also generalizes beyond the experimental data.

Reviewer #3: Xu, Modirshanechi and colleagues present a novel computational framework in which novelty promotes exploration, while surprise adaptively adjusts the model’s rate of adaptation. They show that this model, which uses novelty as a pseudo reward and surprise to adjust a local learning rate, offers a good explanation of human performance on a deep sequential learning task with sparse rewards. In addition, they identify signals correlated with the model’s key variables in recorded EEG. The questions tackled by the authors; how do humans efficiently explore in complex environments with sparse reward information, and how does volatility influence learning and decision making in such an environment, are very much at the edges of both human and machine learning research, positioning this work as highly topical and of interest to a wide audience. However, there are a number of issues that limit the reach of the work presented here. Generally speaking, the computational model is exceptionally complex. I fear that while it offers an explanation of task performance presented here, it relies on several unconventional mechanisms that might obfuscate and limit its generalizability, calling into question its relevance to the more general fields of machine and human learning. I outline some of my specific concerns here:

While it might be a touch trivial, the formulation of novelty does not appear to be consistent with how novelty is commonly framed. Rather, the model relies on a measure of relative stimulus frequency (how rare an image is relative to others). This formulation of novelty feels more akin to forms of surprise than novelty per se. As an example, according to equation 2, an agent that experienced 10000 observations in a world with 2 stimuli, with stimulus A presented 1000 times and B presented 9000 times, the model would treat stimulus A as highly novel despite having observed it 1000 times. Memory research would balk at this. While this is not an issue computationally, as tracking event rarity (be it an unexpected state transition, an unexpected reward, or an unexpected stimulus) likely offers a useful computational variable, it does have implications for the relevant background supporting the notion of ‘novelty’ driven exploration in animals and humans. For example, the authors cite previous work linking dopaminergic activity to the presentation of novel stimuli, but I know of no evidence suggesting that dopamine activity tracks relative stimulus frequency. Perhaps a more traditional formulation of novelty (e.g. a simple monotonically decaying function) could perform just as well?

- Behavioral adaptation is a function of both the learning rate and the magnitude of the prediction error signal. Transitionally, prediction errors offer a means of driving local adaptation (where the prediction errors are large), while the learning rate offers a more global rate of adaptation. From what I understand, the authors use surprise (violation of model-transition expectations) to modulate learning locally at the source of surprise. Could the rapid adaptation at the start of block 2 not also be captured by a larger prediction error than what is offered by the model? What behaviour is captured specifically by a higher learning rate that could not be captured by larger prediction errors? Can a locally modulated learning rate and locally generated prediction error (which may not be accurately modelled) be robustly separated in this task?

- The model uses state frequency as a pseudo reward to guide exploration on episode 1. Relative to a hand-crafted pseudo reward function, it’s refreshing to see a feature of the environment used in this way. However, I fear that a hand crafted pseudo reward has been exchanged for a hand-drafted environment where state frequency is confounded with task performance. Relative frequency is not predictive of reward in many, if not most environments. Does this mechanism for exploration extend to other environments, particularly where rarely encountered states are the only path to the goal (e.g. unique paths of different length that lead to the goal)?

- The cross-validation processes used to optimize free parameters and compare goodness of fit has both pros and cons. One chief benefit comes from the fact that the log-evidence quantifies how well parameter estimates generalize. However, this can be problematic when there are considerably individual differences in task performance as one individual may not in fact be well represented by others in the sample (as appears may be the case here, particularly in the critical first episodes of both blocks). Do model comparison results hold if parameters are optimized and penalized using other methods (more traditional gradient descent optimization and compared using measures like AIC or BIC)? That is, are the results reported here dependent on cross-validation in a relatively small sample?

- The model’s exploration is explicitly turned off once the target image is found. The authors explain that exploration is no longer necessary once the goal is found. However, the task issued to participants was to find the shortest path to the target image. Thus, from the participant’s perspective (since they don’t know the task structure), simply finding the target image does not necessarily solve the task (e.g. there could be more efficient paths to the target image). As participants may continue to search for a shorter path in subsequent episodes, explicitly dampening exploration in models being compared may unfairly hinder their ability to capture variance. Is it necessary that the pseudo-reward mechanism in the model be manually manipulated?

- The model includes a wide array of variables. Are model parameters recoverable - that is - if the model generates a set choice using a specified set of parameters, and the model is then fit to this set of synthetic choice data, are the same parameters recovered? Related to parameter recovery: To what degree do model-based regressors depend on the model parameters - that is - how robust are the parametric regressors used for the EEG analysis robust to perturbations in the parameters themselves?

- The model offering the best fit to human behaviour is a hybrid MB/MF framework. But, it’s not clear to me where the MF and MB components have dissociable preferences. It might help clarify some of the model’s complexity if the contributions made by some of the model’s components were outlined.

- It’s not clear to me why a model-free Q-value scaling parameter is required? Are both systems not learning about the same environmental signals?

- In addition to manually controlling novelty driven exploration, the model also includes manual manipulation of the MB/MF mixture parameter. What is this variation in mixtures capturing in the data?

- Novelty seeking weights are re-applied at the start of block2. However, participants are unaware of the blocking structure - they only experience a surprise when they encounter the unexpected state. Why is novelty seeking re-invigorated for the start of block 2?

- The transformation of Reward and RPE as applied to the EEG analysis deserves some consideration. What does it mean to detect signals correlated with R+RPE and R-RPE when RPE is defined in terms of reward and value expectation? By Reward do the authors mean experienced reward or reward expectation? Are there sufficient trials to perform a robust analysis of Reward with only 12 exposures to the rewarding stimulus?

- The authors run many statistical tests, one for each timepoint in the EEG. I might be mis-interpreting how this was done, but an FDR=0.1 implies that you expect 10% of your tests to be false positives. Does this leave a multiple comparisons issue with models run for each timepoint given the results reported? With 65 measurements, this implies around ~7 false positive tests. It’s hard to tell from the reported data, but it would seem there are roughly that number of positive tests for Surpris, R and RPE.

- The authors orthogonalize novelty, surprise and novelty prediction error to R+ and R-. It would be preferable to allow variables to compete for variance as opposed to explicit orthogonalization.

**Have all data underlying the figures and results presented in the manuscript been provided?**

Reviewer #1: **No: **I didn't see a link to the data. Also, if this moves forward, I think the readers can benefit from a link to a code simulating the winning model.

Reviewer #2: Yes

Reviewer #3: Yes

PLOS authors have the option to publish the peer review history of their article (what does this mean?). If published, this will include your full peer review and any attached files.

Reviewer #1: **Yes: **Nitzan Shahar

Sagol school of Neuroscience

Tel-Aviv University

Reviewer #2: **Yes: **Jan Gläscher

Reviewer #3: No
---

## [Decision Letter · Decision Letter 1]

12 May 2021

Dear Mr. Modirshanechi,

We are pleased to inform you that your manuscript 'Novelty is not Surprise: Human exploratory and adaptive behavior in sequential decision-making' has been provisionally accepted for publication in PLOS Computational Biology. Please note a few minor comments from one reviewer below.

Best regards,

Samuel J. Gershman

Deputy Editor

PLOS Computational Biology

Reviewer's Responses to Questions

**Comments to the Authors:**

Reviewer #1: I believe that all issues were adequately addressed. I have no further comments at this point, and would like to thank you for your hard work in answering our concerns.

Yours,

Nitzan.

Reviewer #2: The authors have address my concerns in their revision. I think the changes in response to all reviewer comment have significantly improved the paper.

There are couple of minor changes to Figure 2 and 7 (and S3 ans S4) that I would like to suggest:

1. The grey lines and data points are really faint and hard to see. Making them a bit darker would probably help.

2. I think it would be informative to show the progress values also for State 7 and State 3 in Blocks 1 and 2 respectively, since these state are also swapped between blocks. In fact, one could also include all progressing states in these subpanels (unless they look mostly like the steadily increasing progress values of State 4.

Pending these minor suggestions I support the publication now and I want to congratulate them on a comprehensive, very interesting, and seminal paper.

**Have the authors made all data and (if applicable) computational code underlying the findings in their manuscript fully available?**

Reviewer #1: **No: **I couldn't find the reference for the shared data.

Reviewer #2: Yes

PLOS authors have the option to publish the peer review history of their article (what does this mean?). If published, this will include your full peer review and any attached files.

Reviewer #1: No

Reviewer #2: No

---

## [Editor Report · Acceptance letter]

1 Jun 2021

PCOMPBIOL-D-21-00152R1

Novelty is not Surprise: Human exploratory and adaptive behavior in sequential decision-making

Dear Dr Modirshanechi,

I am pleased to inform you that your manuscript has been formally accepted for publication in PLOS Computational Biology. Your manuscript is now with our production department and you will be notified of the publication date in due course.

With kind regards,

Katalin Szabo
